# Towards Unbiased Learning in Semi-Supervised Semantic Segmentation

**Rui Sun**[1*]   **Huayu Mai**[1,2*]   **Wangkai Li**[1,2]   **Tianzhu Zhang**[1,2†]
[1]University of Science and Technology of China
[2]National Key Laboratory of Deep Space Exploration, Deep Space Exploration Laboratory
{issunrui, mai556, lwklwk}@mail.ustc.edu.cn, tzzhang@ustc.edu.cn

## ABSTRACT

Semi-supervised semantic segmentation aims to learn from a limited amount of labeled data and a large volume of unlabeled data, which has witnessed impressive progress with the recent advancement of deep neural networks. However, existing methods tend to neglect the fact of class imbalance issues, leading to the Matthew effect, that is, the poorly calibrated model's predictions can be biased towards the majority classes and away from minority classes with fewer samples. In this work, we analyze the Matthew effect present in previous methods that hinder model learning from a discriminative perspective. In light of this background, we integrate generative models into semi-supervised learning, taking advantage of their better class-imbalance tolerance. To this end, we propose DiffMatch to formulate the semi-supervised semantic segmentation task as a conditional discrete data generation problem to alleviate the Matthew effect of discriminative solutions from a generative perspective. Plus, to further reduce the risk of overfitting to the head classes and to increase coverage of the tail class distribution, we mathematically derive a debiased adjustment to adjust the conditional reverse probability towards unbiased predictions during each sampling step. Extensive experimental results across multiple benchmarks, especially in the most limited label scenarios with the most serious class imbalance issues, demonstrate that DiffMatch performs favorably against state-of-the-art methods. Code is available at https://github.com/yuisuen/DiffMatch.

## 1 INTRODUCTION

Machine learning, especially deep learning, has been consistently reported to achieve competitive or even superior performance compared to human beings in certain supervised learning tasks (LeCun et al., 2015). In real-world scenarios, however, its data-driven nature makes it heavily dependent on massive annotations, especially at the dense pixel level, which is laborious and time-consuming to gather (taking semantic segmentation as a case study). To alleviate the data-hunger issue, considerable works (Wang et al., 2023b; Na et al., 2023; Wang et al., 2023a; Liang et al., 2023; Mai et al., 2023; 2025) have turned their attention to semi-supervised semantic segmentation in pursuit of bypassing the labeling cost, demonstrating great potential in widespread applications (Siam et al., 2018; Asgari Taghanaki et al., 2021). Since only limited labeled data is accessible, how to fully exploit a large volume of unlabeled data to improve the model's generalization performance for robust segmentation is thus extremely challenging. To leverage unlabeled data effectively, pseudo-labeling (Lee et al., 2013; Rizve et al., 2021) and consistency regularization (Sajjadi et al., 2016; Laine & Aila, 2016) have emerged as mainstream paradigms for semi-supervised segmentation. Recently, these two paradigms are often assembled in the form of a teacher-student scheme (Wang et al., 2022a; Chen et al., 2023a). In this scheme, the teacher network, with a weakly augmented view, generates pseudo labels to guide the counterparts from the student network in the presence of a strongly augmented view, following the smoothness assumption (Chapelle et al., 2009).

---

*Equal contribution
†Corresponding author

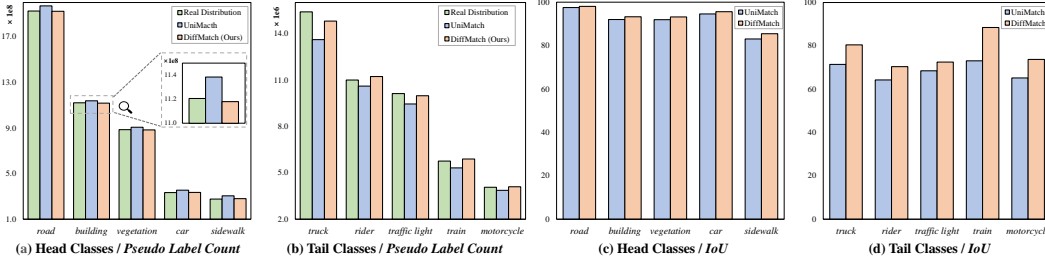

Figure 1: We count the training samples of each class on Cityscapes (Cordts et al., 2016) under 1/16 partition protocols and compare the proposed DiffMatch with the highly competitive Uni-Match (Yang et al., 2022) in terms of *Pseudo Label Count* and *IoU*, assuming that the ground truth for unlabeled data is available solely for theoretical analysis purposes. (a) Prediction distribution of head classes. (b) Prediction distribution of tail classes. (c) Performance of head classes. (d) Performance of tail classes. Our DiffMatch strives to mitigate the Matthew effect raised by the class imbalance issue and stands out for head/tail classes.

From the perspective of probabilistic modeling, almost all de facto methods can be unified as discriminative models, which directly model the conditional probability of discriminating different values across classes for given pixels of an image (*i.e.*, maximizing posterior probability). Despite yielding promising results, these methods tend to neglect the fact of class imbalance issue (*i.e.*, long-tailed distribution). For example, the pixel count of head class *road* can be hundreds of times larger than that of tailed class *motorcycle* in the widely used Cityscapes dataset (Cordts et al., 2016) as shown in Figure 1. This highly skewed distribution can lead to the Matthew effect; that is, the poorly calibrated model's predictions can be *biased* towards the majority classes and away from minority classes with fewer samples. This is a corollary raised by discriminative models, which only learn decision boundaries between classes while disregarding the underlying distribution. In other words, these methods, by minimizing empirical risk under the assumption of low-density separation, are highly fragile to the number of pixels per class (*i.e.*, class imbalance), leading to decision boundaries that can be drastically altered by the majority classes (*i.e.*, confirmation bias (Guo et al., 2017)). This affects the quality of pseudo labels, and then aggressively training with erroneous pseudo labels, in turn, exacerbates the model's bias in a self-reinforcing manner, compromising performance. For example, UniMatch (Yang et al., 2022) tends to prioritize the head classes in Figure 1 (a) over tail classes in Figure 1 (b) in terms of pseudo label count compared to real distribution. To make matters worse, the negative impact is inevitably amplified by inbuilt low-data regimes of semi-supervised segmentation, hindering the learning process. Then, the question naturally arises: *How to effectively alleviate the negative impact raised by class imbalance issue and move towards unbiased learning?*

In this work, we analyze the Matthew effect present in previous methods that hinder the model's learning when dealing with class imbalance issues from a *discriminative perspective*. Compared with the discriminative models, the generative models conceptually exhibit better class-imbalance tolerance, attributed to their better asymptotic error approaching rate (Ng & Jordan, 2001) (detailed in Appendix A). In light of this background, we turn to formulate the semi-supervised semantic segmentation task as a conditional discrete data generation problem to model the underlying distribution, alleviating the Matthew effect of discriminative solutions from a *generative perspective*. To this end, we propose DiffMatch to learn a series of state transitions under the guidance of the input image, transforming noise from a known noise distribution into a prediction that better matches the real distribution, maximizing the mutual information between the learned distribution and the underlying real one. A heuristic explanation of the transition process is that it can be viewed as the human process of discriminating objects, gradually scrutinizing them closer after an initial glance with the naked eye. By formulating the pseudo-label generation of the teacher-student scheme as an optimization problem progressively solved by the denoising diffusion process, DiffMatch favors a better capacity to tackle the severe class imbalance issue in semi-supervised learning. Plus, to further reduce the risk of overfitting to the head classes and to increase coverage of the tail class distribution, we mathematically derive a debiased adjustment based on the state transition function of the diffusion process to adjust the conditional reverse probability towards unbiased predictions during each sampling step. This adjustment, formalized as an additional regularization term, further unlocks the

potential of DiffMatch to mitigate the Matthew effect effectively and is in line with the step-by-step sampling nature of the diffusion model. In practice, tackling class imbalance issue appropriately enables well-calibrated models to generate high-quality pseudo labels (see Figure 1 (c) & (d)), and in turn, improved quality of pseudo labels favorably manifests the mitigation of Matthew effect (see Figure 1 (a) & (b)), moving the learning toward unbiased.

Extensive experiments across diverse benchmarks spanning different backbones demonstrate that our method performs favorably against state-of-the-art semi-supervised semantic segmentation methods, especially in the most limited label scenarios with the most severe class imbalance issues (*e.g.*, +2.6%/+2.0% compared to DDFP (Wang et al., 2024a) and RankMatch (Mai et al., 2024b) respectively on PASCAL *classic* under 1/16 protocol with the ResNet-101), evidencing the merits of modeling underlying distribution in the challenging dense pixel-level classification task.

## 2 RELATED WORK

**Class-Imbalanced Semi-Supervised Segmentation.** Real-world datasets usually yield a class-imbalanced distribution, especially in dense prediction tasks (*e.g.*, semantic segmentation), making the standard training of machine learning models harder to generalize. Existing methods to re-balance the training objective can be roughly categorized into two paradigms: (1) re-sampling based methods (Chawla et al., 2002; He & Garcia, 2009; Byrd & Lipton, 2019; Chang et al., 2021; Shi et al., 2023; Wei et al., 2022) to adjust prediction labels by over-sampling the minority class or under-sampling the majority class. (2) re-weighting based methods (Cao et al., 2019; Cui et al., 2019; Huang et al., 2019; Ren et al., 2018; Hu et al., 2019; Chen et al., 2023d) to influence the loss function conditioned on specific criteria (*e.g.*, imposing the weights by strictly inverse the class frequency). However, all these methods assume all labels are accessible to alleviate the class imbalance issue and thus inapplicable to the unlabelled data in semi-supervised semantic segmentation. Recently, several studies have attempted to transfer these techniques on top of pseudo labels such as re-sampling (Wei et al., 2021), re-weighting (Wang et al., 2022a; Sun et al., 2023c; Xu et al., 2021; He et al., 2021; Wang et al., 2022c; Peng et al., 2023) (*e.g.*, Adsh (Guo & Li, 2022) utilizes adaptive thresholding that can be considered as binary weighting for semi-supervised learning, $U^2PL$ (Wang et al., 2022c) adjusts the threshold adaptively to determine the reliability of pixels and constructs the extra supervised signal based on the negative classes of unreliable pixels, paying more attention to the tail classes), or a combination of both for semi-supervised learning (*e.g.*, AEL (Hu et al., 2021) adaptively balances the training of different categories). Nevertheless, these pseudo labels are often noisy as they are generated from poorly calibrated models. Furthermore, USRN (Guan et al., 2022) explores unbiased subclass regularization for alleviating the class imbalance issue. However, these discriminative methods are still confined to learning decision boundaries, which are brittle to the class imbalance issue, and the inherent nature of contempt for the underlying distribution remains unchanged. As a significant departure from the status quo, we formulate the semi-supervised semantic segmentation task as a conditional discrete data generation problem to model underlying distribution to overcome the shortcomings of discriminative solutions from a generative perspective.

**Diffusion Models for Visual Perception.** In addition to the significant progress in content generation, diffusion models have demonstrated potential for perception tasks (Gu et al., 2022; Chen et al., 2023c; Brempong et al., 2022). Earlier studies primarily investigate latent representations of diffusion models for zero-shot image segmentation (Baranchuk et al., 2021; Burgert et al., 2022) or medical image segmentation (Wolleb et al., 2022; Wu et al., 2022). Despite substantial progress, the outcomes of these efforts remain limited to specific local designs. DiffusionDet (Chen et al., 2023b) and DiffusionInst (Gu et al., 2022) explore diffusion models for query-based object detection (Carion et al., 2020) and instance segmentation (Zhang et al., 2021). Recently, several works have introduced diffusion into various semi-supervised tasks, such as classification, federated learning, time-series classification, and 3d object detection. Among them, both DPT (You et al., 2024) and FedDISC (Yang et al., 2024) aim to introduce an external diffusion model to generate data and utilize these data in a multi-stage training manner. DiffShape (Liu et al., 2024) utilizes diffusion in a self-supervised manner to improve representation capability, and Diffusion-ss3d (Ho et al., 2023) exploits the denoising ability of the diffusion to improve the quality of the pseudo label. However, these methods differ from ours both from motivation to implementation. We comprehensively compare our DiffMatch with these diffusion-based semi-supervised methods in Appendix F. In general, DiffMatch completely utilizes the characteristics of the diffusion process for semi-supervised semantic segmentation, aiming to provide a new perspective to alleviate the Matthew effect.

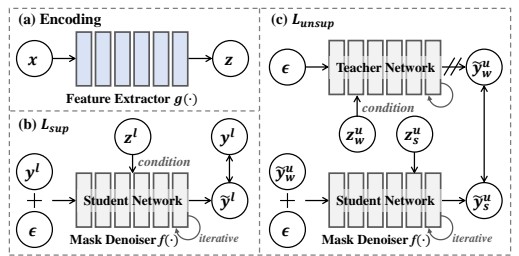

Figure 2: Our DiffMatch framework, which includes a feature extractor $g(\cdot)$ and a mask denoiser $f(\cdot)$. The diffusion process is conducted progressively in mask denoiser, aiming for lightweightness.

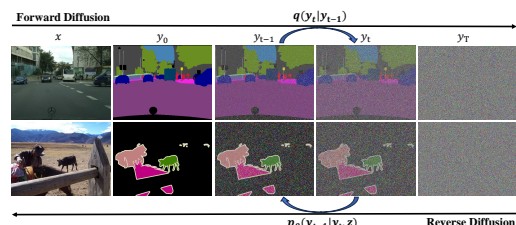

Figure 3: Conditional discrete data generation pipeline for semi-supervised semantic segmentation. Specifically, a conditional diffusion model is employed, where $q$ is the forward diffusion process and $p_\theta$ is the inverse process.

## 3 METHOD

In this section, we first formulate the semi-supervised semantic segmentation problem as preliminaries (Section 3.1), specify the core framework of DiffMatch (Section 3.2 and Section 3.3), and elaborate the training (Section 3.4) and inference (Section 3.5) details.

### 3.1 PROBLEM DEFINITION

Given a labeled set $\mathcal{D}^l = \{(\boldsymbol{x}_i^l, \boldsymbol{y}_i^l)\}_{i=1}^{N^l}$ and an unlabeled set $\mathcal{D}^u = \{\boldsymbol{x}_i^u\}_{i=1}^{N^u}$, where $N^l$ and $N^u$ denote the number of labeled and unlabeled images, respectively, $N^u \gg N^l$, semi-supervised semantic segmentation aims to train a segmentation model with limited labeled data and fully exploit a large volume of unlabeled data. As shown in Figure 2, the popular teacher-student scheme consists of a teacher network and a student network. The student network is guided by two sources of supervision, including the ground truth $\boldsymbol{y}^l$ for the labeled data $\boldsymbol{x}_l$ (yielding supervised loss $\mathcal{L}_{sup}$) and the pseudo labels generated by the teacher network for the unlabeled data (constituting the unsupervised loss $\mathcal{L}_{unsup}$). In specific, for the unlabeled data, the unsupervised loss $\mathcal{L}_{unsup}$ is constructed in the form of consistency regularization, that is, the teacher network with a weakly augmented perturbation view $\boldsymbol{x}_w^u$ generates pseudo labels $\tilde{\boldsymbol{y}}_w^u$ to instruct the counterparts $\tilde{\boldsymbol{y}}_s^u$ from the student network under the presence of a strongly augmented perturbation view $\boldsymbol{x}_s^u$.

The teacher network can either be the same as the student network or an exponentially moving average (EMA) version of the student network. Note that in this paper, the teacher and student networks are identical, following UniMatch (Yang et al., 2022), to ensure simplicity and efficiency. The overall objective is the combination of supervised and unsupervised losses $\mathcal{L} = \mathcal{L}_{sup} + \mathcal{L}_{unsup}$.

In this work, we integrate generative models into semi-supervised learning, taking advantage of its better class-imbalance tolerance. In the next section, we elaborate on the modeling of our DiffMatch in detail, that is, how to realize closer collaboration between the diffusion process and the teacher-student paradigm.

### 3.2 THE DIFFMATCH FRAMEWORK

Figure 2 sheds light on the architecture of generation modeling for proposed DiffMatch. In specific, during training, the Gaussian noise $\boldsymbol{\epsilon}$ controlled by a noise schedule (Ho et al., 2020) is added to the ground truth $\boldsymbol{y}^l$ (from labeled data) or pseudo labels $\tilde{\boldsymbol{y}}_w^u$ (from unlabeled data) to construct the noisy masks. Then, the noisy mask is fused with the pixel embeddings $\boldsymbol{z}$ (acts as the condition) from the feature extractor $g(\cdot)$, and the resulting fused features are fed into a lightweight mask denoiser $f(\cdot)$ to generate the prediction without noise. At the inference phase, DiffMatch generates predictions by reversing the learned diffusion process, which transforms a known Gaussian distribution into a prediction that better matches the real distribution under the guidance of the images, maximizing the mutual information between the learned distribution and the underlying real one.

Due to the iterative nature of the diffusion sampling process, it requires multiple runs of the model during the inference phase. To minimize computational cost, we separate the entire network into two parts: the feature extractor and the mask denoiser following Chen et al. (2023c); Ji et al. (2023).

The former forward only once to extract the pixel embedding, and then the mask denoiser employs it as the condition rather than the raw image to iteratively reads out the prediction mask.

### 3.2.1 THE FEATURE EXTRACTOR

The feature extractor $g(\cdot)$ aims to extract the semantic features of the image $x$ and upsample to a high-resolution pixel embedding $z \in \mathbb{R}^{H \times W \times D}$ with an FPN-style structure (Lin et al., 2017) for sufficient representations. In our experiments, we adopt DeepLabv3+ (Chen et al., 2018) without the last layer classifier for a fair comparison (Na et al., 2023; Sun et al., 2023c), where $D = 256$.

### 3.2.2 THE MASK DENOISER

**Denoiser Network.** The input of the mask decoder $f(\cdot)$ is the concatenation of the noisy mask $y_t$, which is obtained by adding Gaussian noise $\epsilon$ to the ground truth from labeled data ($y^l$) or pseudo labels from unlabeled data ($\tilde{y}_w^u$), and the pixel embedding $z$ from the feature extractor. To further minimize computational cost, we simply stack $L$ layers of deformable attention (Zhu et al., 2020; Ji et al., 2023) as the mask denoiser (The number of layers $L$ is set as 4 by default. Its effect can be referred to in Table 7). This lightweight design enables efficient reuse of shared parameters during multi-step denosing diffusion processes (*i.e.*, after running the feature extractor only once, reuse the efficient denoiser in several iterative steps), while maintaining highly competitive performance. More sophisticated mask denoiser are possible, to leverage recent advances in architecture designs (*e.g.*, TransUNet (Chen et al., 2021a)), but this is not our main focus so we opt for simplicity.

**Forward and Reverse Process.** Inspired by non-equilibrium thermodynamics, the optimization goal of the diffusion model is to maximize the likelihood to favor the alignment of the learned distribution and underlying real one. To this end, the diffusion model learns a series of state transitions (as shown in Figure 3) to transform noise $\epsilon$ (*i.e.*, $y_T = \epsilon$) from a known noise distribution into a data sample $y_0$ from the data distribution $p(y_0)$. To learn this mapping, we first define a forward transition $q(y_t \mid y_{t-1})$ from state $y_{t-1}$ to a more noisy state $y_t$, which is defined as:

$$y_t = \sqrt{\alpha_s} y_{t-1} + \sqrt{1 - \alpha_s}\epsilon \implies q(y_t \mid y_{t-1}) = \mathcal{N}(y_t; \sqrt{\alpha_s} y_{t-1}, (1 - \alpha_s)\mathbf{I}), \quad (1)$$

where $t$ is from uniform density on $[0, 1]$ and $\epsilon$ is drawn from standard normal density. $\alpha_s$ denotes the noise schedule (Ho et al., 2020; Song et al., 2020), meaning that the larger the time step $t$, the more the noise dominates and finally converges to pure Gaussian noise. Denoting the conditional reverse process as $p_\theta(y_t \mid y_{t+1}, z)$, the straightforward objective is:

$$\mathcal{L}_{diff} = \sum_t D_{\text{KL}}[q(y_t \mid y_{t-1}) \| p_\theta(y_t \mid y_{t+1}, z)]. \quad (2)$$

Benefiting from the reparameterization technique, the forward process can be simplified that directly obtaining $y_t$ from $y_0$, as:

$$y_t = \sqrt{\bar{\alpha}_t} y_0 + \sqrt{1 - \bar{\alpha}_t}\epsilon \implies q(y_t \mid y_0) = \mathcal{N}(y_t; \sqrt{\bar{\alpha}_t} y_0, (1 - \bar{\alpha}_t)\mathbf{I}), \quad (3)$$

where $\bar{\alpha}_t = \prod_{s=0}^t \alpha_s$. Similarly, we can learn a mask denoiser $f(\cdot)$ to predict $y_0$ directly from $y_t$ under the guidance of $z$, *i.e.*, $f(y_t, z) = p_\theta(y_0 \mid y_t, z)$. The objective can be simplified to:

$$\mathcal{L}_{diff} = \|f(y_t, z) - y_0\|^2. \quad (4)$$

Note that in our semi-supervised setting, the data samples are either the ground truth mask from labeled data ($y_0 = y^l$) or pseudo labels from unlabeled data ($y_0 = \tilde{y}_w^u$). In specific, deriving from Equation 4, the supervised loss $\mathcal{L}_{sup}$ for labeled data can be formulated as:

$$\mathcal{L}_{sup} = \left\| f(y_t^l, g(x^l)) - y_0^l \right\|^2. \quad (5)$$

In the same way, for the unlabeled data, the unsupervised loss $\mathcal{L}_{unsup}$ can be formulated as:

$$\mathcal{L}_{unsup} = \left\| f(\widetilde{y}_{t,w}^u, g(x_s^u)) - \widetilde{y}_{0,w}^u \right\|^2, \quad (6)$$

where $\widetilde{y}_{0;w}^u = f(\epsilon, g(x_w^u))$ denotes the pseudo labels and $s/w$ means the strong/weak augmentation. Intuitively, the unsupervised loss fits with the consistency regularization of a standard teacher-student paradigm in semi-supervised semantic segmentation. In Algorithm 1, we present the pseudo algorithm of DiffMatch to clearly summarize our method. At this point, we have explored the integration of the diffusion process and the teacher-student paradigm to alleviate the class imbalance issue from a generative perspective.

### 3.3 DEBIASED ADJUSTMENT

Given the long-tailed nature of the class distribution $p(\boldsymbol{y}_0)$ in practice, the learned conditional inverse probability $p_\theta(\boldsymbol{y}_0 \mid \boldsymbol{y}_t, \boldsymbol{z})$ is inevitably biased. To further improve the tolerance of the diffusion model to class imbalance, we propose the debiased adjustment. First, we represent the conditional inverse probability under ideal condition $*$ (*i.e.*, when the class distribution is uniform, $p^*(\boldsymbol{y}_0) = \frac{1}{C}$, where $C$ is the number of classes) as $p_\theta^*(\boldsymbol{y}_0 \mid \boldsymbol{y}_t, \boldsymbol{z})$. With the Bayesian formula, we deduce the relation between $p_\theta(\boldsymbol{y}_0 \mid \boldsymbol{y}_t, \boldsymbol{z})$ and $p_\theta^*(\boldsymbol{y}_0 \mid \boldsymbol{y}_t, \boldsymbol{z})$ (refer to Appendix C for detailed derivation):

$$p_\theta^*\left(\boldsymbol{y}_t \mid \boldsymbol{y}_{t+1}, \boldsymbol{z}\right) = p_\theta\left(\boldsymbol{y}_t \mid \boldsymbol{y}_{t+1}, \boldsymbol{z}\right) \frac{p_\theta\left(\boldsymbol{y}_t\right)}{p_\theta^*\left(\boldsymbol{y}_t\right)} \frac{\hat{q}^*\left(\boldsymbol{y}_{t+1}\right)}{\hat{q}\left(\boldsymbol{y}_{t+1}\right)}. \tag{7}$$

Intuitively, we can obtain the ideal conditional inverse probability $p_\theta^*\left(\boldsymbol{y}_t \mid \boldsymbol{y}_{t+1}, \boldsymbol{z}\right)$ by modulate $p_\theta\left(\boldsymbol{y}_t \mid \boldsymbol{y}_{t+1}, \boldsymbol{z}\right)$ by a factor $\frac{p_\theta(\boldsymbol{y}_t)}{p_\theta^*(\boldsymbol{y}_t)} \frac{\hat{q}^*(\boldsymbol{y}_{t+1})}{\hat{q}(\boldsymbol{y}_{t+1})}$. However, directly estimating this modulation factor at each time step $t$ is highly challenging. Instead, we incorporate it into the training loss function to achieve an equivalent objective. Replacing the $p_\theta(\boldsymbol{y}_0 \mid \boldsymbol{y}_t, \boldsymbol{z})$ in Equation 2 with $p_\theta^*(\boldsymbol{y}_0 \mid \boldsymbol{y}_t, \boldsymbol{z})$:

$$
\begin{aligned}
\mathcal{L}_{diff}^* &= \sum_t D_{\mathrm{KL}}\left[q\left(\boldsymbol{y}_t \mid \boldsymbol{y}_{t-1}\right) \| p_\theta^*\left(\boldsymbol{y}_t \mid \boldsymbol{y}_{t+1}, \boldsymbol{z}\right)\right] \\
&= \sum_t \left\{ D_{\mathrm{KL}}\left[q\left(\boldsymbol{y}_t \mid \boldsymbol{y}_{t-1}\right) \| p_\theta\left(\boldsymbol{y}_t \mid \boldsymbol{y}_{t+1}, \boldsymbol{z}\right)\right] + t D_{\mathrm{KL}}\left[\frac{p_\theta\left(\boldsymbol{y}_{t-1} \mid \boldsymbol{y}_t\right)}{C p_\theta\left(\boldsymbol{y}_0\right)} \| p_\theta^*\left(\boldsymbol{y}_{t-1} \mid \boldsymbol{y}_t\right)\right]\right\} \\
&= \mathcal{L}_{diff} + \sum_t \left\{ t D_{\mathrm{KL}}\left[\frac{p_\theta\left(\boldsymbol{y}_{t-1} \mid \boldsymbol{y}_t\right)}{C p_\theta\left(\boldsymbol{y}_0\right)} \| p_\theta^*\left(\boldsymbol{y}_{t-1} \mid \boldsymbol{y}_t\right)\right]\right\}.
\end{aligned}
\tag{8}
$$

In practice, we approximate the $p_\theta(\boldsymbol{y}_{t-1} \mid \boldsymbol{y}_t)$ with Monte-Carlo sampling from $p_\theta(\boldsymbol{y}_{t-1} \mid \boldsymbol{y}_t, \boldsymbol{z})$ and the loss reduces to:

$$\mathcal{L}_{diff}^* = \left\| f\left(\boldsymbol{y}_t, \boldsymbol{z}\right) - \boldsymbol{y}_0 \right\|^2 + \tau t \left\| f\left(\boldsymbol{y}_t, \boldsymbol{z}\right) - \frac{f\left(\boldsymbol{y}_t, \boldsymbol{z}\right)}{C p\left(\boldsymbol{y}_0\right)} \right\|^2, \tag{9}$$

where $\tau$ is the trade-off weight for the regularization term, set to 0.1 by default, and $C$ is the number of classes. Please refer to Appendix C for detailed derivation. Intuitively, the second term imposes a constraint directly between the prediction of mask denoiser and its roughly debiased version, reducing the risk of overfitting to the head classes and increasing coverage of the tail class distribution. Based on Equation 9, the supervised loss and unsupervised loss are updated as:

$$\mathcal{L}_{sup} = \left\| f\left(\boldsymbol{y}_t^l, g(\boldsymbol{x}^l)\right) - \boldsymbol{y}_0^l \right\|^2 + \tau t \left\| f\left(\boldsymbol{y}_t^l, g(\boldsymbol{x}^l)\right) - \frac{f\left(\boldsymbol{y}_t^l, g(\boldsymbol{x}^l)\right)}{C p\left(\boldsymbol{y}_0^l\right)} \right\|^2, \tag{10}$$

$$\mathcal{L}_{unsup} = \left\| f\left(\widetilde{\boldsymbol{y}}_{t,w}^u, g(\boldsymbol{x}_s^u)\right) - \widetilde{\boldsymbol{y}}_{0,w}^u \right\|^2 + \tau t \left\| f\left(\widetilde{\boldsymbol{y}}_{t,w}^u, g(\boldsymbol{x}_s^u)\right) - \frac{f\left(\widetilde{\boldsymbol{y}}_{t,w}^u, g(\boldsymbol{x}_s^u)\right)}{C p\left(\widetilde{\boldsymbol{y}}_{0,w}^u\right)} \right\|^2. \tag{11}$$

Note that, in our implementation, $p(\boldsymbol{y}_0^l)$ is statistically derived from the ground truth of labeled data while the $p(\widetilde{\boldsymbol{y}}_{0,w}^u)$ is initialized as $p(\boldsymbol{y}_0^l)$ and updated based on its own pseudo label in an exponential moving average (EMA) manner to progressively align the class prior on unlabeled data. By formulating the pseudo label generation of consistency regularization as an optimization problem progressively solved by the denoising diffusion process, DiffMatch bridges the gap by drifting biased prediction towards unbiased learning.

### 3.4 TRAINING

Our main training objective is to learn a series of state transitions under the guidance of input image to transform noise from a known noise distribution into prediction that better matches real class distribution. We adopt analog bits encoding strategy (Chen et al., 2022) to first convert discrete integers from ground truth or pseudo labels into bit strings, and then cast them as real number. When constructing the analog bits, we can shift and scale them into $\{-b, b\}$ (The scaling factor $b$ is by default set to 0.1. Its impact can be referred to in Table 8). To draw samples, we follow the same procedure as sampling in a continuous diffusion model, except that we apply a quantization operation at the end by simply thresholding the generated analog bits. In our implementation, we replace the L2 loss with the standard cross-entropy loss to better suit the segmentation task. The training procedure for the diffusion process is provided in Algorithm 2.

Table 1: Quantitative results of different SSL methods on PASCAL *classic* set. We report mIoU (%) under various partition protocols and show the improvements over *Sup.-only* baseline. The **best** is highlighted in **bold**.

| Method | ResNet-50 | | | | | ResNet-101 | | | | |
|---|---|---|---|---|---|---|---|---|---|---|
| | 1/16 (92) | 1/8 (183) | 1/4 (366) | 1/2 (732) | Full (1464) | 1/16 (92) | 1/8 (183) | 1/4 (366) | 1/2 (732) | Full (1464) |
| *Sup.-only* | 44.0 | 52.3 | 61.7 | 66.7 | 72.9 | 45.1 | 55.3 | 64.8 | 69.7 | 73.5 |
| FixMatch[NeurIPS'20] | 60.1 | 67.3 | 71.4 | 73.7 | 76.9 | 63.9 | 73.0 | 75.5 | 77.8 | 79.2 |
| PCR[NeurIPS'22] | – | – | – | – | – | 70.0 | 74.7 | 77.1 | 78.5 | 80.7 |
| GTA-Seg[NeurIPS'22] | – | – | – | – | – | 70.0 | 73.2 | 75.6 | 78.4 | 80.5 |
| ReCo[ICLR'22] | 64.8 | 72.0 | 73.1 | 74.7 | – | – | – | – | – | – |
| AugSeg[CVPR'23] | 64.2 | 72.1 | 76.1 | 77.4 | 78.8 | 71.0 | 75.4 | 78.8 | 80.3 | 81.3 |
| UniMatch[CVPR'23] | 67.4 | 71.9 | 75.3 | 78.0 | 79.3 | 73.5 | 75.4 | 78.7 | 80.2 | 81.9 |
| NP-SemiSeg[ICML'23] | 65.7 | 72.3 | 75.7 | 77.4 | – | – | – | – | – | – |
| DAW[NeurIPS'23] | 68.5 | 73.1 | 76.3 | 78.6 | 79.7 | 74.8 | 77.4 | 79.5 | 80.6 | 81.5 |
| DDFP[CVPR'24] | – | – | – | – | – | 74.9 | 78.0 | 79.5 | 81.2 | 81.9 |
| RankMatch[CVPR'24] | 71.6 | 74.6 | 76.7 | 78.8 | 80.0 | 75.5 | 77.6 | 79.8 | 80.7 | 82.2 |
| PRCL[IJCV'24] | – | – | – | – | – | 71.2 | 72.2 | 75.2 | 76.2 | 78.3 |
| **DiffMatch (Ours)** | **73.3** | **75.7** | **77.9** | **79.6** | **81.6** | **77.5** | **78.3** | **80.6** | **81.5** | **83.3** |
| Δ ↑ | +29.3 | +23.4 | +16.2 | +12.9 | +8.7 | +32.4 | +23.0 | +15.8 | +11.8 | +9.8 |

Table 2: Quantitative results of different SSL methods on PASCAL *blender* set. We report mIoU (%) under various partition protocols and show the improvements over *Sup.-only* baseline.

| Method | ResNet-50 | | | ResNet-101 | | |
|---|---|---|---|---|---|---|
| | 1/16 (662) | 1/8 (1323) | 1/4 (2646) | 1/16 (662) | 1/8 (1323) | 1/4 (2646) |
| *Sup.-only* | 62.4 | 68.2 | 72.3 | 67.5 | 71.1 | 74.2 |
| FixMatch[NeurIPS'20] | 70.6 | 73.9 | 75.1 | 74.3 | 76.3 | 76.9 |
| AEL[NeurIPS'21] | – | – | – | 77.2 | 77.6 | 78.1 |
| PCR[NeurIPS'22] | – | – | – | 78.6 | 80.7 | 80.8 |
| GTA-Seg[NeurIPS'22] | – | – | – | 77.8 | 80.5 | 80.6 |
| AugSeg[CVPR'23] | 74.7 | 76.0 | 77.2 | 77.0 | 77.3 | 78.8 |
| UniMatch[CVPR'23] | 75.8 | 76.9 | 76.8 | 78.1 | 78.4 | 79.2 |
| CFCG[ICCV'23] | 75.0 | 77.1 | 77.7 | 76.8 | 79.1 | 79.9 |
| NP-SemiSeg[ICML'23] | 73.4 | 76.5 | 76.7 | – | – | – |
| DAW[NeurIPS'23] | 76.2 | 77.6 | 77.4 | 78.5 | 78.9 | 79.6 |
| DDFP[CVPR'24] | – | – | – | 78.3 | 78.8 | 79.8 |
| RankMatch[CVPR'24] | 76.6 | 77.8 | 78.3 | 78.9 | 79.2 | 80.0 |
| PRCL[IJCV'24] | – | – | – | 77.9 | 79.1 | 79.9 |
| **DiffMatch (Ours)** | **77.9** | **78.7** | **79.0** | **80.3** | **81.4** | **81.6** |
| Δ ↑ | +15.5 | +10.5 | +6.7 | +12.8 | +10.3 | +7.4 |

## 3.5 INFERENCE

At the inference phase, the target data sample $y_0$ is reconstructed from noise $y_T$ with the mask denoiser $f(\cdot)$ and an updating rule (Song et al., 2020; Ho et al., 2020) in an iterative Markovian way. We choose the DDIM update rule (Song et al., 2020) for the sampling process. We also represent the trade-off between performance and computation by different sampling steps for multi-step inference in Table 5. Please refer to Algorithm 3 for details about the sampling procedure for diffusion process. Note that to reduce inference overhead, we do not employ any post-processing techniques, such as self-condition (Chen et al., 2022), and sampling drift (Ji et al., 2023), *etc*.

## 4 EXPERIMENTS

In this section, we give comprehensive evaluations of various class-imbalanced datasets. We first describe the experimental setups in Section 4.1. Then, we present the empirical results of our Diff-Match and other compared competitors under extensive setups in Section 4.2. Finally, we present detailed analyses to help understand our method in Section 4.3.

### 4.1 EXPERIMENTAL SETUP

**Datasets.** We conduct experiments on three datasets with severe class-imbalanced issues. (1) **PASCAL VOC 2012** (Everingham et al., 2010) contains 21 classes with 1,464 and 1,449 finely annotated images for training and validation, respectively. We augment the original training set (*i.e.*, *classic*)

Table 3: Quantitative results of different SSL methods on Cityscapes. We report mIoU (%) under various partition protocols and show the improvements over *Sup.-only* baseline.

| Method | ResNet-50 | | | | ResNet-101 | | | |
|---|---|---|---|---|---|---|---|---|
| | 1/16 (186) | 1/8 (372) | 1/4 (744) | 1/2 (1488) | 1/16 (186) | 1/8 (372) | 1/4 (744) | 1/2 (1488) |
| *Sup.-only* | 63.3 | 70.2 | 73.1 | 76.6 | 66.3 | 72.8 | 75.0 | 78.0 |
| FixMatch[NeurIPS'20] | 72.6 | 75.7 | 76.8 | 78.2 | 74.2 | 76.2 | 77.2 | 78.4 |
| AEL[NeurIPS'21] | 74.0 | 75.8 | 76.2 | — | 75.8 | 77.9 | 79.0 | 80.3 |
| PCR[NeurIPS'22] | — | — | — | — | 73.4 | 76.3 | 78.4 | 79.1 |
| GTA-Seg[NeurIPS'22] | — | — | — | — | 69.4 | 72.0 | 76.1 | — |
| AugSeg[CVPR'23] | 73.7 | 76.5 | 78.8 | 79.3 | 75.2 | 77.8 | 79.5 | 80.4 |
| UniMatch[CVPR'23] | 75.0 | 76.8 | 77.5 | 78.6 | 76.6 | 77.9 | 79.2 | 79.5 |
| Co-Train[ICCV'23] | — | 76.3 | 77.1 | — | 75.0 | 77.3 | 78.7 | — |
| NP-SemiSeg[ICML'23] | 73.0 | 77.1 | 78.8 | 78.7 | — | — | — | — |
| DAW[NeurIPS'23] | 75.2 | 77.5 | 79.1 | 79.5 | 76.6 | 78.4 | 79.8 | 80.6 |
| DDFP[CVPR'24] | — | — | — | — | 77.1 | 78.1 | 79.8 | 80.8 |
| RankMatch[CVPR'24] | 75.4 | 77.7 | 79.2 | 79.5 | 77.1 | 78.6 | 80.0 | 80.7 |
| PRCL[IJCV'24] | — | — | — | — | 73.4 | 77.0 | 77.9 | 80.0 |
| **DiffMatch (Ours)** | **76.5** | **78.3** | **79.8** | **80.0** | **77.8** | **79.1** | **80.5** | **81.3** |
| Δ ↑ | **+13.2** | **+8.1** | **+6.7** | **+3.4** | **+11.5** | **+6.3** | **+5.5** | **+3.3** |

with additional 9,118 coarsely annotated images in SBD (Hariharan et al., 2011) to get a *blender* training set following other researches (Chen et al., 2021b; Hu et al., 2021). According to statistics, the pixel number of the head class *background* is more than $200\times$ that of the tail class *bicycle*. (2) **Cityscapes** (Cordts et al., 2016) consists of 2,975 images for training and 500 images for validation with 19 classes. The ratio of head class *road* to tail class *motorcycle* reaches 400. (3) **COCO** (Lin et al., 2014), composed of 118k/5k training/validation images, is a more severe class-imbalanced dataset, containing 81 classes to predict, with over $10,000$ head-to-tail ratio.

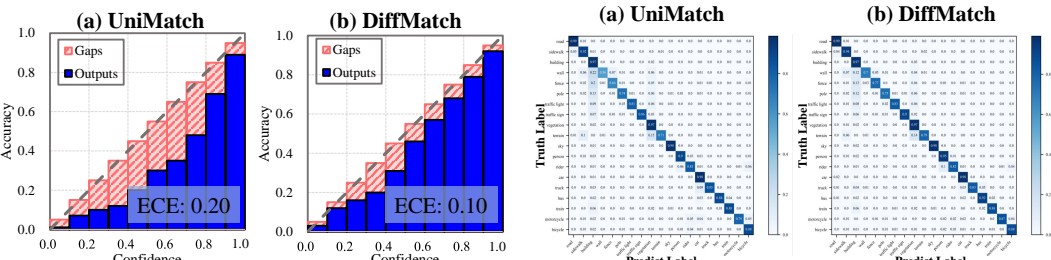

Figure 4: Calibration on unlabeled data produced by UniMatch (left) and DiffMatch (right).

Figure 5: Confusion matrix on unlabeled data of UniMatch (left) and DiffMatch (right).

**Implementation Details.** For a fair and exhaustive comparison, we use ResNet-50/101 (He et al., 2016) pretrained on ImageNet (Krizhevsky et al., 2012) and Xception-65 (Chollet, 2017) as the backbones and DeepLabv3+ (Chen et al., 2018) as the decoder. We set the sampling step as 3 at inference, the number of layers in mask denoiser $L$ as 4 and the scaling factor $b$ as 0.1 for all experiments. During training, we randomly crop $513 \times 513$ for PASCAL and COCO datasets, and train 80 and 30 epochs, respectively. For Cityscapes, the cropsize is set as $801 \times 801$ and the training epoch is 240. The batch size of the three datasets is set to 8. Polynomial Decay learning rate policy is applied throughout the whole training. The strong augmentation contains feature dropout, random color jitter, grayscale and Gaussian blur. The weak augmentation consists of random crop, resize and horizontal flip. All experiments are conducted on $8\times$ RTX 3090 GPUs (memory is 24G/GPU).

## 4.2 EMPIRICAL RESULTS

We evaluate our method on PASCAL (*classic* and *blender*), Cityscapes datasets with ResNet-50/101, and COCO dataset with Xception-65 under different semi-supervised learning settings (*i.e.*, partition protocols). The partition protocol (*e.g.*, 1/16) indicates the ratio of labeled data used in training to the entire training set. It is worth noting that the smaller the partition protocol, the less labeled data is used for training, and the more biased the training may be. The consistently dominant performance under all partition protocols with different backbones on all datasets against other competi-

tors (FixMatch (Sohn et al., 2020), PseudoSeg (Zou et al., 2020), AEL (Hu et al., 2021), ReCo (Liu et al., 2021), PC2Seg (Zhong et al., 2021), PCR (Xu et al., 2022), GTA-Seg (Jin et al., 2022), UniMatch (Yang et al., 2022), AugSeg (Zhao et al., 2023c), NP-SemiSeg (Wang et al., 2023c), DAW (Sun et al., 2023c), CFCG (Li et al., 2023a), Co-Train (Li et al., 2023b), MKD (Yuan et al., 2023), DDFP (Wang et al., 2024a), RankMatch (Mai et al., 2024b), PRCL (Xie et al., 2024)) proves the effectiveness of our DiffMatch against the class imbalance issue, evidencing the merits of modeling underlying distribution in the challenging dense pixel-level classification task.

**Results on PASCAL.** Table 1 and Table 2 show the comparison of our method with the SOTA methods on PASCAL *classic* and *blender* set. Compared with the supervised-only (*Sup.-only*) model, our method achieves considerable performance improvements, demonstrating that the information in unlabeled data is effectively utilized in our method. Moreover, in the label-scarce scenario, *e.g.*, 1/16 (92) in PASCAL *classic*, our approach achieves $73.3\%$ and $77.5\%$ mIoU with the backbone ResNet-50 and ResNet-101, boosting the SOTA DAW (Sun et al., 2023c) by $4.8\%$ and $2.7\%$, respectively. These superior results prove that our training is more unbiased.

**Results on Cityscapes.** Table 3 summarizes the performance of our DiffMatch and compared methods on the Cityscapes dataset. For the more class-imbalanced dataset (the ratio of head class road to tail class motorcycle reaches $400$), our method still achieves SOTA performance. Specifically, compared with the leading methods DAW (Sun et al., 2023c), DiffMatch improves up to 1.3%/1.2% at absolute mIoU gain under 1/16 partition protocols with ResNet-50/ResNet-101, respectively, showing the superiority of our method over discriminative methods.

**Results on COCO.** COCO is a large-scale dataset where the class imbalance issue is most severe (the number of head-to-tail ratio is more than 10,000). In Table 4, DiffMatch achieves surprising performance lift compared with the discriminative model. For example, under the 1/512 partition protocol, the performance of DiffMatch is superior to that of UniMatch (Yang et al., 2022) (34.6% *vs.* 31.9%), this is in line with the goal of DiffMatch against class imbalance issue.

## 4.3 DETAILED ANALYSES

**Performance in Head&Tail Classes.** Considering that the Matthew effect refers to the bias in model predictions, it can also be viewed as a measure of model calibration. This directly impacts the quality of pseudo-labels for unlabeled data, thereby affecting the model's performance across different classes. Therefore, we compare DiffMatch to other competitive methods to analyze the effectiveness of addressing class imbalance by examining the performance of head/tail classes. To show the source of our absolute performance gain, we present the mIoU of the top-5 classes ($\text{mIoU}_h$) and the bottom-5 classes ($\text{mIoU}_t$) under PASCAL *classic* 1/16 (92) with ResNet-50. To make a comprehensive comparison, we reproduce several methods under the same experiment setting, including class-imbalanced learning methods ♦, SSL methods ♠, and recently proposed class-imbalanced SSL methods ♣. (1) Specifically, for class-imbalanced learning, we consider the two most popular paradigms: a) Re-Sampling (Byrd & Lipton, 2019) and b) Re-weighting (Cui et al., 2019); (2) for SSL methods, we take Fixmatch (Sohn et al., 2020), ReCo (Liu et al., 2021) and NP-SemiSeg (Wang et al., 2023c) into consideration. (3) To further show the efficacy of our proposal, we also compare it with the recently proposed algorithms that consider SSL and class imbalance issues simultaneously, including DARP (Kim et al., 2020), CReST (Wei et al., 2021), FreeMatch (Wang et al., 2022a), DARS (He et al., 2021), AEL (Hu et al., 2021), $\text{U}^2\text{PL}$ (Wang et al., 2022c) and USRN (Guan et al., 2022). Please refer to Section 2 for more details.

As depicted in Table 6, we have the following findings: (1) It is not desirable to directly apply the class-imbalanced learning method to SSL tasks because it does not utilize unlabeled data. (2) SSL methods achieve certain performance gains, but still underperform in the tail classes. (3) Thanks to the modeling of distributions and the derived debiased adjustment, DiffMatch yields favorable performance especially in the tail classes, effectively alleviating the Matthew effect. To better understand the prediction bias of each class, as Figure 6 illustrates, DiffMatch achieves more unbiased predictions on all 21 classes. Moreover, we provide training curves for the number of pseudo labels in the head (*road*) and tail (*motorcycle*) classes in the Appendix D, demonstrating the effectiveness of our DiffMatch in mitigating the Matthew effect.

**Accuracy *vs.* Efficiency.** We show the dynamic trade-off of DiffMatch between accuracy and efficiency in Table 5. To begin with, we construct a discriminative baseline (*Dis.* Baseline) with the

Table 4: Quantitative results of different SSL methods on COCO.

| Method | 1/512 | 1/256 | 1/128 | 1/64 | 1/32 |
|---|---|---|---|---|---|
| *Sup.-only* | 22.9 | 28.0 | 33.6 | 37.8 | 42.2 |
| PseudoSeg | 29.8 | 37.1 | 39.1 | 41.8 | 43.6 |
| PC2Seg | 29.9 | 37.5 | 40.1 | 43.7 | 46.1 |
| MKD | 30.2 | 38.0 | 42.3 | 45.5 | 47.3 |
| UniMatch | 31.9 | 38.9 | 44.4 | 48.2 | 49.8 |
| **DiffMatch (Ours)** | **34.6** | **41.9** | **47.2** | **49.8** | **52.4** |
| Δ ↑ | +11.7 | +13.9 | +13.6 | +12.0 | +10.2 |

Table 5: Accuracy *vs.* Efficiency.

| Method | | mIoU (92) | mIoU (1464) | FPS (↑) | #Param |
|---|---|---|---|---|---|
| UniMatch | | 67.4 | 79.3 | 24.9 | 40.5M |
| *Dis.* Baseline | | 67.9 | 79.5 | – | 44.9M |
| DiffMatch w/o *adj.* | | 72.2 | 81.3 | – | 44.9M |
| **DiffMatch** | step1 | 68.7 | 79.9 | 23.3 | |
| | step2 | 71.2 | 80.7 | 21.2 | |
| | *step3* | **73.3** | **81.6** | 19.6 | 44.9M |
| | step4 | 73.3 | 81.4 | 18.2 | |
| | step5 | 73.4 | 81.7 | 16.9 | |

Table 6: Performance of head & tail classes.

| ResNet-50 | PASCAL *classic* 1/16 (92) | | |
|---|---|---|---|
| | mIoU | mIoU$_h$ | mIoU$_t$ |
| *Sup.-only* | 44.0 | 66.5 | 28.1 |
| ♦ Re-Sampling | 45.6 | 67.8 | 29.3 |
| ♦ Re-weighting | 46.2 | 68.3 | 30.1 |
| ♠ FixMatch | 60.1 | 78.4 | 48.4 |
| ♠ ReCo | 64.8 | 81.2 | 49.6 |
| ♠ NP-SemiSeg | 65.8 | 82.7 | 50.2 |
| ♣ DARP | 61.5 | 79.9 | 49.0 |
| ♣ CReST | 62.2 | 80.6 | 49.4 |
| ♣ FreeMatch | 62.3 | 80.2 | 49.1 |
| ♣ DARS | 62.7 | 82.5 | 50.3 |
| ♣ AEL | 66.3 | 84.2 | 51.1 |
| ♣ U$^2$PL | 67.4 | 85.3 | 53.7 |
| ♣ USRN | 66.8 | 83.9 | 51.8 |
| **DiffMatch (Ours)** | **73.3** | **89.3** | **66.8** |

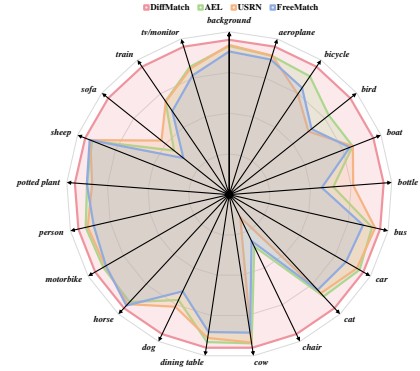

Figure 6: Normalized performance on PASCAL *classic* 92 for per class.

same extra deformable attention layers. (1) Comparing the 1st and 2nd rows, we can see that simply increasing the number of parameters in the model does not lead to an effective performance improvement. Then, 2nd *vs.* 3rd indicates that the performance improvement of DiffMatch primarily stems from modeling the underlying distribution, as opposed to discriminative models (*Dis.* Baseline). (2) Comparing the 3nd row (DiffMatch w/o *adj.*) and the final DiffMatch, we can observe a clear performance lift credited to debiased adjustment. This suggests the effectiveness of debiased adjustment to adjust the conditional reverse probability, reducing the risk of overfitting to the head classes and increasing coverage of the tail class distribution. (3) With the sampling step increase, the performance gets better (same result can also be observed in Figure 10). When adopting 3 sampling steps, the performance is further boosted while maintaining comparable FPS. These results show that DiffMatch can iteratively infer multiple times with reasonable time cost.

**Quality of Pseudo Label.** To take a close look at DiffMatch, we showcase the confusion matrix (Figure 5) and expected calibration error (Figure 4) on unlabeled data to directly measure the performance of different models in the Matthew effect and model calibration respectively, on Cityscapes 1/16 partition. The results show that the raw pseudo-labels generated by UniMatch are biased toward the major classes. For example, there are more than 20% examples that belong to class *wall* are predicted wrongly as class *building*. On the contrary, our DiffMatch can achieve a more unbiased confusion matrix, striving to mitigate the Matthew effect. These results indicate that the quality of pseudo-labels is actually improved, which can help to improve the generalization performance. Similarly, a better-calibrated model is obtained thanks to the modeling of the underlying distribution by DiffMatch (Figure 4). Based on this, well-calibrated models will generate high-quality pseudo labels, and in turn, improved quality of pseudo labels could result in a better distribution estimation.

## 5  CONCLUSION

In this paper, we analyze the Matthew effect in previous methods that hinder model learning when dealing with class imbalance issues from a discriminative view. we propose DiffMatch to formulate the semi-supervised semantic segmentation task as a conditional discrete data generation problem to model underlying distribution against the Matthew effect. DiffMatch offers a fresh generative perspective to alleviating class imbalance, and we believe it has the potential to complement other semi-supervised learning strategies to facilitate future advancements.

## ACKNOWLEDGMENTS

This work was partially supported by the National Defense Basic Scientific Research Program (Grant JCKY2021601B013).

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

## A  MOTIVATION FOR GENERATIVE MODELS TO ALLEVIATE CLASS IMBALANCE ISSUE

Previous research (Ng & Jordan, 2001) theoretically derived the differences in generalization error $\varepsilon(\cdot)$ between discriminative (Dis) and generative models (Gen) under ideal conditions (*i.e.*, with an infinite number of samples $\infty$), where $m$ denotes the number of samples, $n$ is the number of model parameters, and $G(\cdot)$ represents a small meaningful bound.

$$\varepsilon\left(h_{\text{Dis}}\right) \leq \varepsilon\left(h_{\text{Dis},\infty}\right) + O\left(\sqrt{\frac{n}{m}\log\frac{m}{n}}\right) \tag{12}$$

$$\varepsilon\left(h_{\text{Gen}}\right) \leq \varepsilon\left(h_{\text{Gen},\infty}\right) + G\left(O\left(\sqrt{\frac{1}{m}\log n}\right)\right) \tag{13}$$

The above theory demonstrates that the *asymptotic error approaching rate* of generative models is $O(\log n)$, which is better than the discriminative model's $(O(n))$. In other words, under the same number of model parameters, generative models can approach the optimal form under the ideal condition (*i.e.*, infinite training sample) with fewer training samples (logarithmic number, *i.e.*, $O(\log n)$), compared to the discriminative model, which requires a linear number of samples $(O(n))$. This provides a special bonus for the inherent class imbalance problem in semi-supervised semantic segmentation, particularly for tail classes. Specifically, generative models have better potential to enable tail classes with extremely limited sample quantity to converge to the form assumed under sufficient sample conditions, conceptually bridging the gap with the ample samples of head classes, *i.e.*, *better class-imbalance tolerance*.

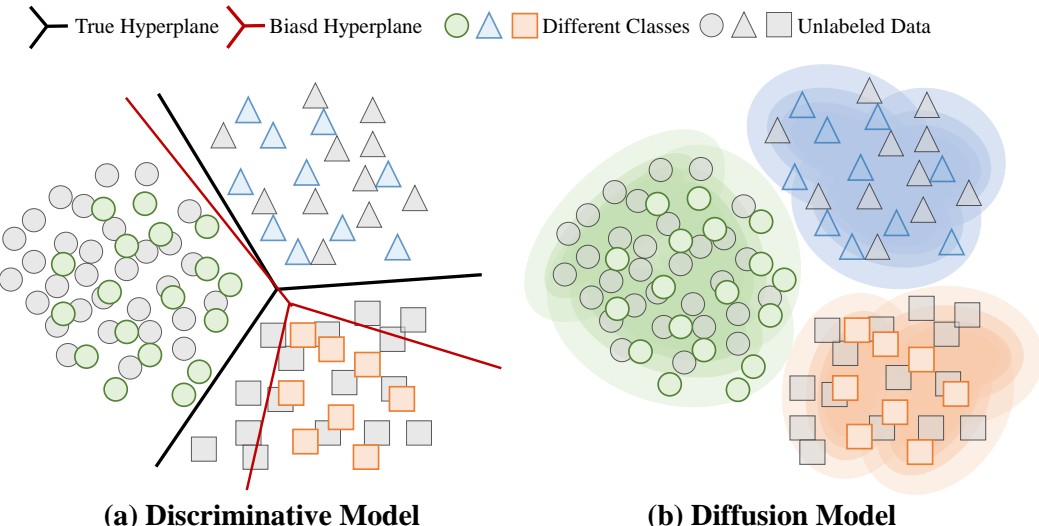

**(a) Discriminative Model**      **(b) Diffusion Model**

Figure 7: Illustration on discriminative model *vs.* diffusion model.

On the other hand, from the perspective of optimization objectives, diffusion-based generative models and discriminative models have fundamentally different optimization objectives. Specifically, discriminative models are typically trained by minimizing empirical risk, aiming to minimize the prediction error or loss function of the model solely on the training data. In this case, these methods, only learning decision boundaries between classes, are highly fragile to the number of pixels per class (*i.e.*, class imbalance), leading to decision boundaries that can be drastically altered by the majority classes (as shown in the left part of Figure 7). In contrast, diffusion-based generative models use log-likelihood as their optimization objective, maximizing the log-likelihood between the explicitly modeled class distribution and the underlying real one (as shown in the right part of Figure 7). Benefiting from modeling probabilistic density, diffusion-based generative models pay more attention to the class distribution rather than the boundaries across classes. Therefore, they conceptually exhibit *better tolerance to class imbalance*.

# B PSEUDO ALGORITHM

In this section, we summarize the pseudo algorithm of DiffMatch in Algorithm 1. The inputs consist of a labeled set $\mathcal{D}^l = \{(\boldsymbol{x}_i^l, \boldsymbol{y}_i^l)\}_{i=1}^{N^l}$ and an unlabeled set $\mathcal{D}^u = \{\boldsymbol{x}_i^u\}_{i=1}^{N^u}$, where $N^u \gg N^l$. The feature extractor $g(\cdot)$, mask denoiser $f(\cdot)$, weak augmentation $\omega$, strong augmentation $s$, and time step $t$ are defined. The algorithm iterates over each batch of labeled data $(\boldsymbol{x}^l, \boldsymbol{y}^l)$ and unlabeled data $\boldsymbol{x}^u$. For labeled data, the pixel embedding $\boldsymbol{z}^l$ is extracted using $g(\cdot)$, noise is injected into $\boldsymbol{y}^l$ to obtain $\boldsymbol{y}_t^l$ via the forward process (Equation 3), and the noisy mask $\boldsymbol{y}_t^l$ is denoised conditioned on $\boldsymbol{z}^l$ and $t$ using $f(\cdot)$ in the reverse process. The supervised loss $\mathcal{L}_{sup}$ is calculated by Equation 10. For unlabeled data, weak and strong augmentations are applied on $\boldsymbol{x}^u$ to obtain $\boldsymbol{x}_w^u$ and $\boldsymbol{x}_s^u$ respectively. Their pixel embeddings $\boldsymbol{z}_w^u$ and $\boldsymbol{z}_s^u$ are extracted using $g(\cdot)$. The pseudo label $\widetilde{\boldsymbol{y}}_{0,w}^u$ is obtained by denoising $\boldsymbol{\epsilon}$ conditioned on $\boldsymbol{z}_w^u$ in the reverse process. Noise is injected into $\widetilde{\boldsymbol{y}}_{0,w}^u$ to obtain $\widetilde{\boldsymbol{y}}_{t,w}^u$ via the forward process, and the noisy mask $\widetilde{\boldsymbol{y}}_{t,w}^u$ is denoised conditioned on $\boldsymbol{z}_s^u$ using $f(\cdot)$ in the reverse process. The unsupervised loss $\mathcal{L}_{unsup}$ is calculated by Equation 11. Finally, the model is updated by performing gradient backward on $\mathcal{L}_{sup} + \mathcal{L}_{unsup}$.

---

**Algorithm 1** Pseudo algorithms of DiffMatch.

---

1: **Inputs:** Labeled Set $\mathcal{D}^l = \{(\boldsymbol{x}_i^l, \boldsymbol{y}_i^l)\}_{i=1}^{N^l}$, Unlabeled Set $\mathcal{D}^u = \{\boldsymbol{x}_i^u\}_{i=1}^{N^u}$ $(N^u \gg N^l)$

2: **Define:** Feature Extractor $g(\cdot)$, Mask Denoiser $f(\cdot)$, Weak Augmentation $w$, Strong Augmentation $s$, time step $t$

3: **Output:** Feature Extractor $g(\cdot)$, Mask Denoiser $f(\cdot)$

4: **for** each batch of $(\boldsymbol{x}^l, \boldsymbol{y}^l)$, $\boldsymbol{x}^u$ in $\mathcal{D}_l$, $\mathcal{D}_u$ **do**

5:     *# Labeled Data:*

6:     Extract pixel embedding $\boldsymbol{z}^l$ for $\boldsymbol{x}^l$ using $g(\cdot)$

7:     Inject noise into $\boldsymbol{y}^l$ and obtain $\boldsymbol{y}_t^l$ by Equation 3          ▷ *Forward Process*

8:     Denoise the noisy mask $\boldsymbol{y}_t^l$ conditioned on $\boldsymbol{z}^l$ and $t$ using $f(\cdot)$          ▷ *Reverse Process*

9:     Calculate $\mathcal{L}_{sup}$ by Equation 10          ▷ *Supervised Loss*

10:     *# Unlabeled Data:*

11:     Obtain $\boldsymbol{x}_w^u$ and $\boldsymbol{x}_s^u$ by applying weak and strong augmentation on $\boldsymbol{x}^u$, respectively

12:     Extract pixel embedding $\boldsymbol{z}_w^u$ and $\boldsymbol{z}_s^u$ using $g(\cdot)$

13:     Obtain the pseudo label $\widetilde{\boldsymbol{y}}_{0,w}^u$ by denoising $\boldsymbol{\epsilon}$ conditioned on $\boldsymbol{z}_w^u$ using $f(\cdot)$

14:               ▷ *Reverse Process*

15:     Inject noise into $\widetilde{\boldsymbol{y}}_{0,w}^u$ and obtain $\widetilde{\boldsymbol{y}}_{t,w}^u$ by Equation 3          ▷ *Forward Process*

16:     Denoise the noisy mask $\widetilde{\boldsymbol{y}}_{t,w}^u$ conditioned on $\boldsymbol{z}_s^u$ using $f(\cdot)$          ▷ *Reverse Process*

17:     Calculate $\mathcal{L}_{unsup}$ by Equation 11          ▷ *Unsupervised Loss*

18:     Gradient backward $\mathcal{L}_{sup} + \mathcal{L}_{unsup}$          ▷ *Update Model*

19: **end for**

---

**Algorithm 2** Diffusion Training Process

```python
def alpha_cumprod(t, ns=0.0002, ds=0.00025):
  """cosine noise schedule"""
  n = torch.cos((t + ns) / (1 + ds) * math.pi / 2) ** -2
  return -torch.log(n - 1, eps=1e-5)

def train(images, masks):
  """images: [b, 3, h, w], masks: [b, 1, h, w]"""
  img_enc = feature_extractor(images) # encode image
  mask_enc = encoding(masks) # encode gt or pseudo labels
  mask_enc = (sigmoid(mask_enc) * 2 - 1) * scale # corrupt gt or pseudo
      labels
  eps = uniform(0, 1), normal(mean=0, std=1)
  mask_crpt = sqrt(alpha_cumprod(t)) * mask_enc + sqrt(1 - alpha_cumprod(t
      )) * eps
  # predict and backward
  mask_pred = mask_denoiser(mask_crpt, mask_enc, t)
  loss = CE_loss(mask_pred, masks) # calculate the loss after debiased
      adjustment
  return loss
```

**Algorithm 3** Diffusion Sampling Process

```
def ddim(mask_t, mask_pred, t_now, t_next):
  """ estimate x at t_next with DDIM update rule"""
  α_now = alpha_cumprod(t_now)
  α_next = alpha_cumprod(t_next)
  mask_enc = encoding(mask_pred)
  mask_enc = (sigmoid(mask_enc) * 2 - 1) * scale
  eps = 1/√(1-α_now) * (mask_t - √(α_now) * mask_enc)
  mask_next = √(α_next) * x_pred + √(1-α_now) * eps
  return mask_next

def sample(images, steps, td=1):
  """steps: sample steps, td: time difference"""
  img_enc = feature_extractor(images)
  mask_t = normal(0, 1) # [b, 256, h/4, w/4]
  for step in range(steps):
    # time intervals
    t_now = 1 - step / steps
    t_next = max(1 - (step + 1 + td) / steps, 0)
    # predict mask_0 from mask_t
    mask_pred = mask_denoiser(mask_t, img_enc, t_now)
    # estimate mask_t at t_next
    mask_t = ddim(mask_t, mask_pred, t_now, t_next)
  return mask_pred
```

## C    DERIVATION OF $\mathcal{L}^*_{diff}$

Here, we present the detailed derivation of $\mathcal{L}^*_{diff}$ from the learning of the diffusion model. Denoting the underlying conditional distribution as $\hat{q}$, we can rewrite the conditional reverse probability $\hat{q}(\boldsymbol{y}_t \mid \boldsymbol{y}_{t+1}, \boldsymbol{z})$ according to Bayes' formula following Dhariwal & Nichol (2021):

$$
\begin{aligned}
\hat{q}(\boldsymbol{y}_t \mid \boldsymbol{y}_{t+1}, \boldsymbol{z}) &= \frac{q(\boldsymbol{y}_t \mid \boldsymbol{y}_{t+1}) \hat{q}(\boldsymbol{z} \mid \boldsymbol{y}_t)}{\hat{q}(\boldsymbol{z} \mid \boldsymbol{y}_{t+1})} \\
&= \frac{q(\boldsymbol{y}_t \mid \boldsymbol{y}_{t+1}) \hat{q}(\boldsymbol{y}_t \mid \boldsymbol{z}) \hat{q}(\boldsymbol{z}) \hat{q}(\boldsymbol{y}_{t+1})}{\hat{q}(\boldsymbol{y}_{t+1} \mid \boldsymbol{z}) \hat{q}(\boldsymbol{z}) \hat{q}(\boldsymbol{y}_t)} \\
&= \frac{q(\boldsymbol{y}_t \mid \boldsymbol{y}_{t+1}) \hat{q}(\boldsymbol{y}_t \mid \boldsymbol{z}) \hat{q}(\boldsymbol{y}_{t+1})}{\hat{q}(\boldsymbol{y}_{t+1} \mid \boldsymbol{z}) \hat{q}(\boldsymbol{y}_t)}.
\end{aligned}
\tag{14}
$$

Since the conditional diffusion model is trained to fit a prior distribution with known conditions by definition, we can approximate $\hat{q}(\boldsymbol{y}_t)$ with $p_\theta(\boldsymbol{y}_t)$ and have:

$$
p_\theta(\boldsymbol{y}_t \mid \boldsymbol{y}_{t+1}, \boldsymbol{z}) = \frac{q(\boldsymbol{y}_t \mid \boldsymbol{y}_{t+1}) \hat{q}(\boldsymbol{y}_t \mid \boldsymbol{z}) \hat{q}(\boldsymbol{y}_{t+1})}{\hat{q}(\boldsymbol{y}_{t+1} \mid \boldsymbol{z}) p_\theta(\boldsymbol{y}_t)}.
\tag{15}
$$

Given the long tailed nature of the class distribution $p(\boldsymbol{y}_0)$ in practice, the learned conditional inverse probability $p_\theta(\boldsymbol{y}_t \mid \boldsymbol{y}_{t+1}, \boldsymbol{z})$ is inevitably biased. To further reduce the risk of overfitting to the head classes and to increase coverage of the tail class distribution, we propose the debiased adjustment. First, we represent the conditional inverse probability under ideal condition (*i.e.*, when the class distribution is uniform, $p^*(\boldsymbol{y}_0) = \frac{1}{C}$, where $C$ is the number of classes) as $p_\theta^*(\boldsymbol{y}_0 \mid \boldsymbol{y}_t, \boldsymbol{z})$. In the same way:

$$
p_\theta^*(\boldsymbol{y}_t \mid \boldsymbol{y}_{t+1}, \boldsymbol{z}) = \frac{q(\boldsymbol{y}_t \mid \boldsymbol{y}_{t+1}) \hat{q}^*(\boldsymbol{y}_t \mid \boldsymbol{z}) \hat{q}^*(\boldsymbol{y}_{t+1})}{\hat{q}^*(\boldsymbol{y}_{t+1} \mid \boldsymbol{z}) p_\theta^*(\boldsymbol{y}_t)}.
\tag{16}
$$

Since $\boldsymbol{y}_0$ is uniquely determined by $\boldsymbol{z}$, we have:

$$
\hat{q}^*(\boldsymbol{y}_t \mid \boldsymbol{z}) = \hat{q}^*(\boldsymbol{y}_t \mid \boldsymbol{y}_0) \overset{\textcircled{\scriptsize 1}}{=} \hat{q}(\boldsymbol{y}_t \mid \boldsymbol{y}_0) = \hat{q}(\boldsymbol{y}_t \mid \boldsymbol{z}),
\tag{17}
$$

where the equality ① holds because $\hat{q}^*(\boldsymbol{y}_t \mid \boldsymbol{y}_0)/\hat{q}(\boldsymbol{y}_t \mid \boldsymbol{y}_0)$ is conditioned on $\boldsymbol{y}_0$, *i.e.*, unrelated to $p(\boldsymbol{y}_0)$. In the same way:

$$
\hat{q}^*(\boldsymbol{y}_{t+1} \mid \boldsymbol{z}) = \hat{q}(\boldsymbol{y}_{t+1} \mid \boldsymbol{z}).
\tag{18}
$$

In other words, $\hat{q}^*(\boldsymbol{y}_t \mid \boldsymbol{z})$ and $\hat{q}^*(\boldsymbol{y}_{t+1} \mid \boldsymbol{z})$ are not affected by the class distribution. Combining the above equations, we have:

$$
p_\theta^*(\boldsymbol{y}_t \mid \boldsymbol{y}_{t+1}, \boldsymbol{z}) = p_\theta(\boldsymbol{y}_t \mid \boldsymbol{y}_{t+1}, \boldsymbol{z}) \frac{p_\theta(\boldsymbol{y}_t)}{p_\theta^*(\boldsymbol{y}_t)} \frac{\hat{q}^*(\boldsymbol{y}_{t+1})}{\hat{q}(\boldsymbol{y}_{t+1})}.
\tag{19}
$$

It can be seen that there is only a factor of difference (*i.e.*, $\frac{p_\theta(\boldsymbol{y}_t)}{p_\theta^*(\boldsymbol{y}_t)} \frac{\hat{q}^*(\boldsymbol{y}_{t+1})}{\hat{q}(\boldsymbol{y}_{t+1})}$) between the ideal conditional inverse process $p_\theta^*(\boldsymbol{y}_t \mid \boldsymbol{y}_{t+1}, \boldsymbol{z})$ and the actual conditional inverse process $p_\theta(\boldsymbol{y}_t \mid \boldsymbol{y}_{t+1}, \boldsymbol{z})$. However, the factor is difficult to obtain directly. Therefore, We convert it into the training loss and gradually remove this difference during training. Since $\frac{\hat{q}^*(\boldsymbol{y}_{t+1})}{\hat{q}(\boldsymbol{y}_{t+1})}$ is independent of the model parameters, it follows from Menon et al. (2020) that the sign should be reversed when converting the post-hoc adjustment factors into the training loss, giving us:

$$
p_\theta^*(\boldsymbol{y}_t \mid \boldsymbol{y}_{t+1}, \boldsymbol{z}) = p_\theta(\boldsymbol{y}_t \mid \boldsymbol{y}_{t+1}, \boldsymbol{z}) \frac{p_\theta^*(\boldsymbol{y}_t)}{p_\theta(\boldsymbol{y}_t)}.
\tag{20}
$$

Then we get the unbiased loss for the conditional diffusion model by replacing the $p_\theta(\boldsymbol{y}_0 \mid \boldsymbol{y}_t, \boldsymbol{z})$ in Equation 2 with $p_\theta^*(\boldsymbol{y}_0 \mid \boldsymbol{y}_t, \boldsymbol{z})$:

$$
\begin{aligned}
\mathcal{L}^*_{diff} &= \sum_t D_{\mathrm{KL}} \left[ q \left( \boldsymbol{y}_t \mid \boldsymbol{y}_{t-1} \right) \| p^*_\theta \left( \boldsymbol{y}_t \mid \boldsymbol{y}_{t+1}, \boldsymbol{z} \right) \right] \\
&= \sum_t \mathbb{E}_q \left[ -\log \frac{p^*_\theta \left( \boldsymbol{y}_t \mid \boldsymbol{y}_{t+1}, \boldsymbol{z} \right)}{q \left( \boldsymbol{y}_t \mid \boldsymbol{y}_{t-1} \right)} \right] \\
&= \sum_t \left\{ \mathbb{E}_q \left[ -\log \frac{p_\theta \left( \boldsymbol{y}_t \mid \boldsymbol{y}_{t+1}, \boldsymbol{z} \right)}{q \left( \boldsymbol{y}_t \mid \boldsymbol{y}_{t-1} \right)} \right] + \mathbb{E}_q \left[ -\log \frac{p^*_\theta \left( \boldsymbol{y}_t \right)}{p_\theta \left( \boldsymbol{y}_t \right)} \right] \right\} \\
&= \sum_t \left\{ D_{\mathrm{KL}} \left[ q \left( \boldsymbol{y}_t \mid \boldsymbol{y}_{t-1} \right) \| p_\theta \left( \boldsymbol{y}_t \mid \boldsymbol{y}_{t+1}, \boldsymbol{z} \right) \right] + \mathbb{E}_q \left[ -\log \frac{p^*_\theta \left( \boldsymbol{y}_t \right)}{p_\theta \left( \boldsymbol{y}_t \right)} \right] \right\}.
\end{aligned} \tag{21}
$$

Focus on the second item of the above equation:

$$
\begin{aligned}
&\sum_t \mathbb{E}_q \left[ -\log \frac{p^*_\theta \left( \boldsymbol{y}_t \right)}{p_\theta \left( \boldsymbol{y}_t \right)} \right] \\
&= \sum_t \mathbb{E}_q \left\{ -\log \mathbb{E}_{p_\theta} \left[ \frac{p^*_\theta \left( \boldsymbol{y}_0 \right) \prod_{t'=1}^t p^*_\theta \left( \boldsymbol{y}_{t'} \mid \boldsymbol{y}_{t'-1} \right)}{p_\theta \left( \boldsymbol{y}_0 \right) \prod_{t'=1}^t p_\theta \left( \boldsymbol{y}_{t'} \mid \boldsymbol{y}_{t'-1} \right)} \right] \right\} \\
&= \sum_t \mathbb{E}_q \left\{ -\log \mathbb{E}_{p_\theta} \left[ \frac{p^*_\theta \left( \boldsymbol{y}_0 \right) \prod_{t'=1}^t p^*_\theta \left( \boldsymbol{y}_{t'-1} \mid \boldsymbol{y}_{t'} \right) \frac{p^*_\theta(\boldsymbol{y}_{t'})}{p^*_\theta(\boldsymbol{y}_{t'-1})}}{p_\theta \left( \boldsymbol{y}_0 \right) \prod_{t'=1}^t p_\theta \left( \boldsymbol{y}_{t'-1} \mid \boldsymbol{y}_{t'} \right) \frac{p_\theta(\boldsymbol{y}_{t'})}{p_\theta(\boldsymbol{y}_{t'-1})}} \right] \right\} \\
&\overset{②}{\leqslant} \sum_t \mathbb{E}_q \left\{ \mathbb{E}_{p_\theta} \left[ -\log \frac{p^*_\theta \left( \boldsymbol{y}_0 \right) \prod_{t'=1}^t p^*_\theta \left( \boldsymbol{y}_{t'-1} \mid \boldsymbol{y}_{t'} \right) \frac{p^*_\theta(\boldsymbol{y}_{t'})}{p^*_\theta(\boldsymbol{y}_{t'-1})}}{p_\theta \left( \boldsymbol{y}_0 \right) \prod_{t'=1}^t p_\theta \left( \boldsymbol{y}_{t'-1} \mid \boldsymbol{y}_{t'} \right) \frac{p_\theta(\boldsymbol{y}_{t'})}{p_\theta(\boldsymbol{y}_{t'-1})}} \right] \right\} \\
&= \sum_t \mathbb{E}_q \left\{ \mathbb{E}_{p_\theta} \left[ \sum_{t'=1}^t -\log \frac{p_\theta \left( \boldsymbol{y}_0 \right) p^*_\theta \left( \boldsymbol{y}_{t'-1} \mid \boldsymbol{y}_{t'} \right)}{p^*_\theta \left( \boldsymbol{y}_0 \right) p_\theta \left( \boldsymbol{y}_{t'-1} \mid \boldsymbol{y}_{t'} \right)} \right] \right\} \\
&= \mathbb{E}_q \left\{ \mathbb{E}_{p_\theta} \left[ \sum_t \sum_{t'=1}^t -\log \frac{p_\theta \left( \boldsymbol{y}_0 \right) p^*_\theta \left( \boldsymbol{y}_{t'-1} \mid \boldsymbol{y}_{t'} \right)}{p^*_\theta \left( \boldsymbol{y}_0 \right) p_\theta \left( \boldsymbol{y}_{t'-1} \mid \boldsymbol{y}_{t'} \right)} \right] \right\} \\
&\overset{③}{=} \sum_t t \mathbb{E}_q \left[ -\log \frac{p^*_\theta \left( \boldsymbol{y}_{t'-1} \mid \boldsymbol{y}_{t'} \right)}{\frac{p_\theta \left( \boldsymbol{y}_{t'-1} \mid \boldsymbol{y}_{t'} \right)}{C p_\theta \left( \boldsymbol{y}_0 \right)}} \right] \\
&= \sum_t t D_{\mathrm{KL}} \left[ \frac{p_\theta \left( \boldsymbol{y}_{t-1} \mid \boldsymbol{y}_t \right)}{C p_\theta \left( \boldsymbol{y}_0 \right)} \| p^*_\theta \left( \boldsymbol{y}_{t-1} \mid \boldsymbol{y}_t \right) \right],
\end{aligned} \tag{22}
$$

where the inequality ② holds due to Jensen's Inequality, and the equality ③ is holds because $p^* (\boldsymbol{y}_0) = \frac{1}{C}$. In practice, we approximate the $p_\theta(\boldsymbol{y}_{t-1} \mid \boldsymbol{y}_t)$ with Monte-Carlo samples from $p_\theta(\boldsymbol{y}_{t-1} \mid \boldsymbol{y}_t, \boldsymbol{z})$ and the loss reduce to:

$$
\begin{aligned}
\mathcal{L}^*_{diff} &= \| f \left( \boldsymbol{y}_t, \boldsymbol{z} \right) - \boldsymbol{y}_0 \|^2 + \tau t \left\| f \left( \boldsymbol{y}_t, \boldsymbol{z} \right) - \frac{f \left( \boldsymbol{y}_t, \boldsymbol{z} \right)}{C p \left( \boldsymbol{y}_0 \right)} \right\|^2 \\
&= \mathcal{L}_{diff} + \tau t \left\| f \left( \boldsymbol{y}_t, \boldsymbol{z} \right) - \frac{f \left( \boldsymbol{y}_t, \boldsymbol{z} \right)}{C p \left( \boldsymbol{y}_0 \right)} \right\|^2.
\end{aligned} \tag{23}
$$

# D   Training Curve for Head&Tail Classes

Figure 8 provides a comparative analysis of the training samples of the head class *road* and the tail class *motorcycle* on the Cityscapes Cordts et al. (2016) under the 1/16 partition protocol as the training progresses. The proposed DiffMatch is compared with the highly competitive UniMatch Yang et al. (2022) in terms of pseudo label count, assuming that the ground truth for unlabeled data is available solely for theoretical analysis purposes.

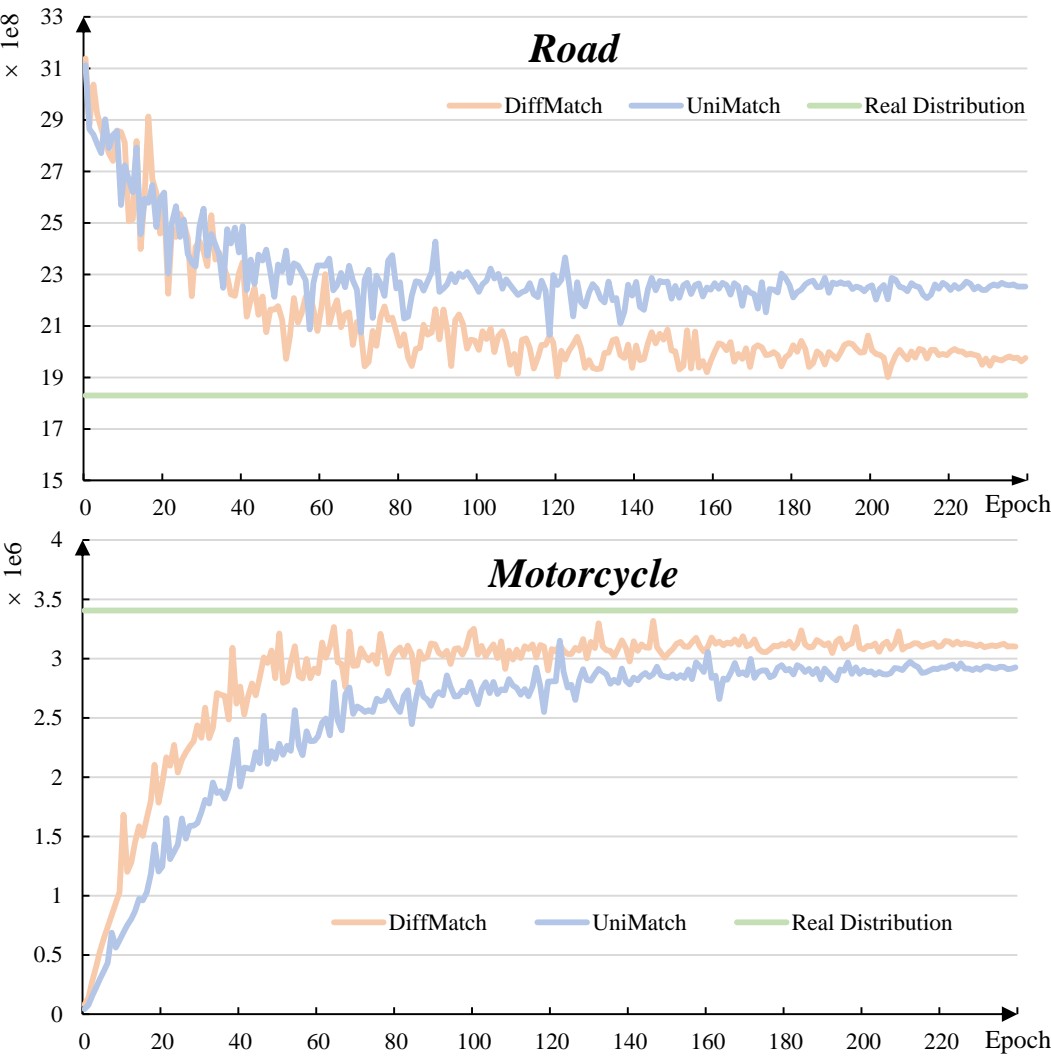

Figure 8: We count the training samples of **head class** *road* and **tail class** *motorcycle* on Cityscapes (Cordts et al., 2016) under 1/16 partition protocols as the training processes, and compare the proposed DiffMatch with the highly competitive UniMatch (Yang et al., 2022) in terms of *Pseudo Label Count*, assuming that the ground truth for unlabeled data is available solely for theoretical analysis purposes. Our DiffMatch strives to mitigate the Matthew effect raised by the class imbalance issue and stands out for head/tail classes.

The top plot in Figure 8 illustrates the prediction distribution of the head class *road*. UniMatch generates a significantly higher pseudo label count compared to the real distribution, indicating its tendency to over-predict the dominant class. In contrast, DiffMatch exhibits a pseudo label count that is more aligned with the real distribution, demonstrating its ability to mitigate the bias towards the head class. The bottom plot in Figure 8 depicts the prediction distribution of the tail class *motorcycle*. UniMatch generates substantially fewer pseudo labels compared to the real distribution, highlighting

its under-prediction of the minority class. Conversely, DiffMatch demonstrates a pseudo label count that is much closer to the real distribution, showcasing its effectiveness in addressing the under-representation of the tail class.

The comparative analysis in Figure 8 substantiates the effectiveness of DiffMatch in leveraging the advantages of generative models to alleviate the Matthew effect. By incorporating a diffusion model and theoretically deriving a debiased adjustment (3.3), DiffMatch effectively mitigates the bias towards head classes and the under-prediction of tail classes, promoting unbiased learning in semi-supervised semantic segmentation. This finding aligns with the quantitative results analyzed in the *"Performance in Head&Tail Classes"* of Section 4.3; please refer to it for more details.

## E    DETAILED ANALYSES OF HYPER-PARAMETERS

**Decoder Depth.** Table 7 investigates the effect of the decoder depth, *i.e.*, the number of layers in the mask denoiser $f(\cdot)$. The results demonstrate that increasing the depth initially improves the model accuracy, with the optimal performance achieved at 4 layers (73.3% mIoU on PASCAL *classic* 1/16(92)). However, further increasing the depth beyond 4 layers leads to the saturation of performance. This observation aligns with the goal behind the lightweight design of the mask denoiser, which enables efficient reuse of shared parameters during multi-step denoising diffusion processes while maintaining highly competitive performance. The chosen architecture with 4 layers strikes a balance between accuracy and efficiency, with a parameter count of 44.9M.

Table 7: Evaluation of number of layers

| #Layer $L$ | mIoU(92) | mIoU(1464) | #Param |
|:---:|:---:|:---:|:---:|
| 1 | 70.1 | 79.5 | 42.4M |
| 2 | 71.2 | 80.6 | 43.3M |
| *4* | **73.3** | **81.6** | 44.9M |
| 6 | 72.6 | 80.8 | 45.8M |
| 12 | 71.9 | 81.1 | 49.9M |

**Scaling Factor.** Table 8 explores the impact of the scaling factor $b$ used in the analog bits encoding strategy (Section 3.4). The scaling factor determines the range $\{-b, b\}$ into which the analog bits are shifted and scaled. The results show that a suitable scaling factor is necessary for optimal performance. As the scaling factor increases, the model accuracy improves until reaching a peak at $b = 0.1$ (73.3% mIoU on PASCAL *classic* 1/16(92) and 81.6% mIoU on PASCAL *classic* Full(1464)). Further increasing the scaling factor leads to a decline in performance. We hypothesize that a larger scaling factor retains more easy cases with the same time step, potentially affecting the balance between easy and hard cases during training.

Table 8: Evaluation of scaling factor.

| Scale $b$ | mIoU(92) | mIoU(1464) |
|:---:|:---:|:---:|
| 0.01 | 71.7 | 80.9 |
| 0.05 | 72.2 | 81.2 |
| *0.1* | **73.3** | **81.6** |
| 0.2 | 70.7 | 80.8 |
| 0.5 | 70.6 | 80.5 |

**Regularization Term $\tau$.** Table 9 examines the influence of the trade-off weight $\tau$ for the regularization term in the debiased adjustment. The regularization term imposes a constraint between the prediction of mask denoiser and its roughly debiased version, reducing the risk of overfitting to head classes and increasing coverage of tail class distribution. The results indicate that setting $\tau = 0.1$ yields the optimal performance, that is, 73.3% mIoU on PASCAL *classic* 1/16(92) and 81.6% mIoU on PASCAL *classic* Full(1464).

Table 9: Evaluation of trade-off weight for the regularization term $\tau$.

| $\tau$ | mIoU(92) | mIoU(1464) |
|---|---|---|
| 0.01 | 71.8 | 80.0 |
| 0.02 | 72.2 | 80.4 |
| 0.05 | 72.7 | 80.9 |
| *0.1* | **73.3** | **81.6** |
| 0.2 | 71.9 | 80.2 |

## F    COMPARISON WITH OTHER DIFFUSION-BASED SEMI-SUPERVISED METHODS

As diffusion gains popularity in visual perception, researchers have introduced it into various semi-supervised tasks (You et al., 2024; Yang et al., 2024; Liu et al., 2024; Ho et al., 2023), such as classification, federated learning, time-series classification and 3d object detection. In the following, we will comprehensively and meticulously compare our DiffMatch with these diffusion-based semi-supervised methods and summarize in Table 10 to highlight the originality of our work.

Different from our DiffMatch, both DPT (You et al., 2024) and FedDISC (Yang et al., 2024) *utilize an external diffusion model* to generate additional data and demonstrate their effectiveness in facilitating the original model training. Specifically, DPT introduces a from-scratch diffusion-based conditional generative model to address the scarcity of labeled data in semi-supervised classification task in three stages: train the original classifier on limited labeled data to predict pseudo-labels; train the conditional generative model using these pseudo-labels to generate labeled data; retrain the classifier with a combination of limited real and vast generated labeled data. FedDISC addresses the challenge of semi-supervised federated learning by introducing a well-trained diffusion model. To alleviate the communication burden between the server and clients, the diffusion model generates rich client-style data for the server, conditioned on the cluster centroid of client data representations, thereby facilitating model training on the server.

Regarding DiffShape (Liu et al., 2024), although it explores integrating the diffusion process into semi-supervised time-series classification, it does so through a self-supervised mechanism *rather than incorporating it into the teacher-student network paradigm*. Specifically, DiffShape employs large amounts of unlabeled instance subsequences as conditions in the diffusion process to generate the subsequences themselves, enhancing similarity in the generated sequences compared to the original ones, thereby improving representation capability in a self-supervised manner.

For Diffusion-ss3d (Ho et al., 2023), although it integrates the diffusion process into the teacher-student network paradigm in semi-supervised 3D object detection, we categorize it as *a noise-to-filter paradigm*, leveraging the denoising capability of diffusion models to generate more accurate 3D bounding boxes as pseudo labels. Specifically, Diffusion-ss3d first predicts coarse bounding boxes (fixed bounding box candidate points) with a detection model, which can be considered as intermediate states in the diffusion process, and then employs the diffusion model as a denoising process to obtain other parameters of the bounding box (*e.g.*, bounding box size). Overall, this paradigm *partially exploits the characteristics of the diffusion process*, that is, the denoising ability, to improve the quality of the bounding boxes prediction.

Distinguished from these methods, Our DiffMatch integrates the diffusion process into the teacher-student network for semi-supervised semantic segmentation, which can be viewed as *a noise-to-prediction paradigm*. Motivated by the potential of generative models with better tolerance to class imbalance, our DiffMatch *learns the complete process* of transforming noise from a known distribution to class predictions (all states from time 0 to time T). Additionally, we *mathematically derive a debiased adjustment based on the state transition function* encapsulated in the diffusion process to further mitigate the Matthew effect. This mathematical formulation translates into strong empirical performance on real-world datasets, particularly in scenarios with the most limited labeled data and the most severe class imbalance. In general, DiffMatch completely utilizes the characteristics of the diffusion process in a different problem for semi-supervised semantic segmentation, aiming to provide a new perspective to alleviate the Matthew effect.

Table 10: Comparison with other diffusion-based semi-supervised methods.

| | DPT | FedDISC | DiffShape | Diffusion-ss3d | DiffMatch (Ours) |
|---|---|---|---|---|---|
| **Task** | **classification** | **federated learning** | **time-series classification** | **3d object detection** | **semantic segmentation** |
| **Motivation** | harnessing the **data generation capability** of Diffusion to **alleviate data scarcity** | harnessing the **data generation capability** of Diffusion to **alleviate data scarcity** | using diffusion in a **self-supervised manner** to **improve representation capability** | exploiting the **denoising ability** of Diffusion to **improve the quality of pseudo label** | leveraging the **well class-imbalance tolerance** of Diffusion to alleviate the **Matthew effect** |
| **Implementation** | introducing a **from-scratch external diffusion model** | introducing a **well-trained external diffusion model** | integrating the diffusion process through a **self-supervised mechanism** | integrating the diffusion process into the teacher-student framework in a **noise-to-filter paradigm** | integrating the diffusion process into the teacher-student framework in a **noise-to-prediction paradigm** |
| **Note** | | | | learning an **incomplete** diffusion process | (1) learning a **complete** diffusion process (2) **mathematically deriving a debiased adjustment** based on the state transition function |

## G    LIMITATION AND SOCIETY IMPACT

DiffMatch may face a potential limitation in terms of increased computational cost during multi-step inference. And how to adapt the number of inference steps to the degree of change in the generation state is a feasible direction. Within this paper, we present an approach for semi-supervised semantic segmentation, a pivotal research area in the realm of computer vision, with no apparent negative societal implications known thus far.

## H    EXTENDED DISCUSSION ON RELATED WORK

**Semi-Supervised Segmentation.** Semantic segmentation has achieved conspicuous achievements attributed to the recent advances in the deep neural network (Mai et al., 2024a; Sun et al., 2023d). Among them, semi-supervised semantic segmentation is a fundamental task with extensive applications in scene understanding (Mittal et al., 2019; Wu et al., 2023; Wang et al., 2024b; 2023d;e; 2022b), medical image analysis (Yu et al., 2019; Bai et al., 2023; Zhao et al., 2025; Chi et al., 2024; Sun et al., 2021), brain neuroscience (Sun et al., 2023b;a; Chen et al., 2025; Luo et al., 2024; Pan et al., 2023), autonomous driving (Li et al., 2024b; Pan et al., 2024; Li et al., 2024a) and remote sensing interpretation (Wang et al., 2021; Bandara & Patel, 2022; Yuan et al., 2024). These algorithms leverage the mature combination of pseudo-labeling and consistency regularization (Lai et al., 2021; Zhong et al., 2021; Ouali et al., 2020; Chen & Lian, 2022) to improve performance. More recently, UniMatch (Yang et al., 2022) acknowledges the characteristics of semantic segmentation and incorporates appropriate data augmentations into FixMatch (Sohn et al., 2020), resulting in a concise yet powerful semi-supervised semantic segmentation baseline. Subsequently, a series of works aim to improve segmentation performance mainly in the following aspects. (1) Employ reasonable augmentation strategies to expand the augmentation space. For example, AugSeg (Zhao et al., 2023c) increases the randomness in RandAugment (Cubuk et al., 2020) for richer data augmentation space. iMAS (Zhao et al., 2023b) employs adaptive augmentations and supervisions conditioned on the model state. (2) Design effective teacher networks for better guidance. For example, Switch (Na et al., 2023) targets the coupling problem in the exponentially moving average (EMA) update process of teacher-student network and proposes a dual-teacher structure in an ensemble manner. (3) Utilize external knowledge to enhance the quality of pseudo labels. For example, LOGIC (Liang et al., 2023) integrates symbolic reasoning derived from symbolic knowledge to mitigate erroneous pseudo labels. SemiVL (Hoyer et al., 2025) incorporates a CLIP encoder (Radford et al., 2021), pre-trained on large-scale data, into semi-supervised semantic segmentation and employs a language-aware decoder to introduce text modality priors. (4) Enhance consistency regularization (Sun et al., 2024; Howlader et al., 2025b) to effectively exploit the information contained in unlabeled data. For example, RankMatch (Mai et al., 2024b) utilizes inter-pixel correlations to construct more safe and effective supervision signals, which are in line with the nature of semantic segmentation. MPMC (Howlader et al., 2025a) identifies the classes present in an image region to incorporate pixel-level contextual information, thereby exploring more supervision signals. Despite yielding promising results, these methods tend to neglect the fact of class imbalance issue. In this paper, we strive to alleviate the negative impact (Matthew effect) raised by class imbalance issue and move towards unbiased semi-supervised learning.

**Class-Imbalanced Semi-Supervised Segmentation.** Real-world datasets usually yield a class-imbalanced distribution, especially in dense prediction tasks (*e.g.*, semantic segmentation), making the standard training of machine learning models harder to generalize. Existing methods to re-balance the training objective can be roughly categorized into two paradigms: (1) Re-sampling based methods (Chawla et al., 2002; He & Garcia, 2009; Byrd & Lipton, 2019; Chang et al., 2021; Shi et al., 2023; Wei et al., 2022) attempt to artificially balance the training data distribution. These approaches either employ over-sampling techniques to increase the representation of minority classes or utilize under-sampling strategies to reduce the dominance of majority classes. While effective in certain scenarios, these methods often struggle with the trade-off between maintaining data diversity and achieving balanced class distributions. (2) Re-weighting based methods (Cao et al., 2019; Cui et al., 2019; Huang et al., 2019; Ren et al., 2018; Hu et al., 2019; Chen et al., 2023d) focus on modifying the loss function to prioritize learning from under-represented classes. These approaches typically assign importance weights to different classes based on various criteria, such as inverse class frequency or dynamic class-wise difficulty measures. Although these methods have shown

promising results, they often require careful tuning of weighting schemes to prevent instability during training. However, all these methods assume all labels are accessible to alleviate the class imbalance issue and thus inapplicable to the unlabelled data in semi-supervised semantic segmentation. Recently, several studies have attempted to transfer these techniques on top of pseudo labels such as re-sampling (Wei et al., 2021), re-weighting (Wang et al., 2022a; Sun et al., 2023c; Xu et al., 2021; He et al., 2021; Wang et al., 2022c; Peng et al., 2023) (*e.g.*, Adsh (Guo & Li, 2022) utilizes adaptive thresholding that can be considered as binary weighting for semi-supervised learning, $U^2PL$ (Wang et al., 2022c) adjusts the threshold adaptively to determine the reliability of pixels and constructs the extra supervised signal based on the negative classes of unreliable pixels, paying more attention to the tail classes), or a combination of both for semi-supervised learning (*e.g.*, AEL (Hu et al., 2021) adaptively balances the training of different categories). Nevertheless, these pseudo labels are often noisy as they are generated from poorly calibrated models. Furthermore, USRN (Guan et al., 2022) explores unbiased subclass regularization for alleviating the class imbalance issue. However, these discriminative methods are still confined to learning decision boundaries, which are brittle to the class imbalance issue, and the inherent nature of contempt for the underlying distribution remains unchanged. As a significant departure from the status quo, we formulate the semi-supervised semantic segmentation task as a conditional discrete data generation problem to model underlying distribution to overcome the shortcomings of discriminative solutions from a generative perspective.

**Diffusion Models for Visual Perception.** In addition to the significant progress in content generation, diffusion models have demonstrated incremental potential in the domain of perception (Chen et al., 2023b; Gu et al., 2022; Chen et al., 2023c; Brempong et al., 2022). Earlier studies primarily delve into investigating latent representations of diffusion models for zero-shot image segmentation (Baranchuk et al., 2021; Burgert et al., 2022) or applied diffusion models to medical image segmentation (Wolleb et al., 2022; Wu et al., 2022). Despite substantial progress, the outcomes of these efforts remain limited to specific local designs. The recent Pix2Seq-D (Chen et al., 2023c) extends the bit-diffusion (Chen et al., 2022) to panoptic segmentation, marking the first work of such expansion in a broader context. Additionally, DiffusionDet (Chen et al., 2023b) and Diffusion-Inst (Gu et al., 2022) explore diffusion models for query-based object detection (Carion et al., 2020) and instance segmentation (Zhang et al., 2021). Most recently, groundbreaking work has extended the application of diffusion models to a comprehensive range of dense visual perception tasks (Ji et al., 2023; Zhao et al., 2023a; Zheng et al., 2024). These latest developments have achieved promising results across multiple challenging scenarios, further solidifying the position of diffusion models as a versatile and powerful tool in the visual perception domain. Recently, several works have introduced diffusion into various semi-supervised tasks, such as classification, federated learning, time-series classification, and 3d object detection. Among them, both DPT (You et al., 2024) and FedDISC (Yang et al., 2024) aim to introduce an external diffusion model to generate data and utilize these data in a multi-stage training manner. DiffShape (Liu et al., 2024) utilizes diffusion in a self-supervised manner to improve representation capability, and Diffusion-ss3d (Ho et al., 2023) exploits the denoising ability of the diffusion to improve the quality of the pseudo label. However, these methods differ from ours both from motivation to implementation. We comprehensively and meticulously compare our DiffMatch with these diffusion-based semi-supervised methods in Appendix F. In general, DiffMatch completely utilizes the characteristics of the diffusion process for semi-supervised semantic segmentation, aiming to provide a new perspective to alleviate the Matthew effect.

## I  MORE VISUALIZATION

Here, we provide additional visualizations to qualitatively assess the performance of DiffMatch in comparison to other methods. Figure 9 showcases the segmentation results on the PASCAL VOC dataset, highlighting the effectiveness of DiffMatch in obtaining more accurate semantic segmentation, particularly for pixels that are incorrectly segmented as the most dominant class by other methods. For example, in the 2nd row, FreeMatch, UniMatch, and RankMatch encounter difficulties in accurately segmenting the *person* pixels. They misclassify a considerable portion of the *person* pixels as the *horse* class. These misclassifications can be attributed to the class imbalance issue, where the models are inclined to favor the majority classes, resulting in subpar segmentation performance for the less represented classes like *person*. In contrast, DiffMatch demonstrates a notable ability to overcome these challenges and generate more precise segmentations. By incorporating a genera-

tive perspective and employing a debiased adjustment, DiffMatch effectively mitigates the Matthew effect stemming from class imbalance. As a result, it accurately segments the *person* pixels.

Furthermore, Figure 10 offers additional insights into the inference trajectory of DiffMatch across different diffusion sampling steps. The ground truth segmentation is provided as a reference, and the segmentation results at steps 1, 2, and 3 are visualized. As the number of sampling steps increases, the segmentation quality progressively improves, with finer details and more accurate boundary delineation.

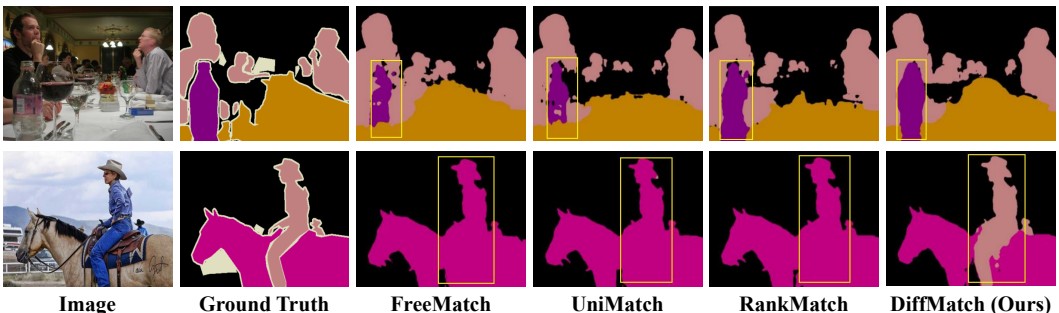

Figure 9: Qualitative results on PASCAL VOC dataset. DiffMatch can obtain more accurate segmentation for pixels that are inaccurately segmented as the most dominant class.

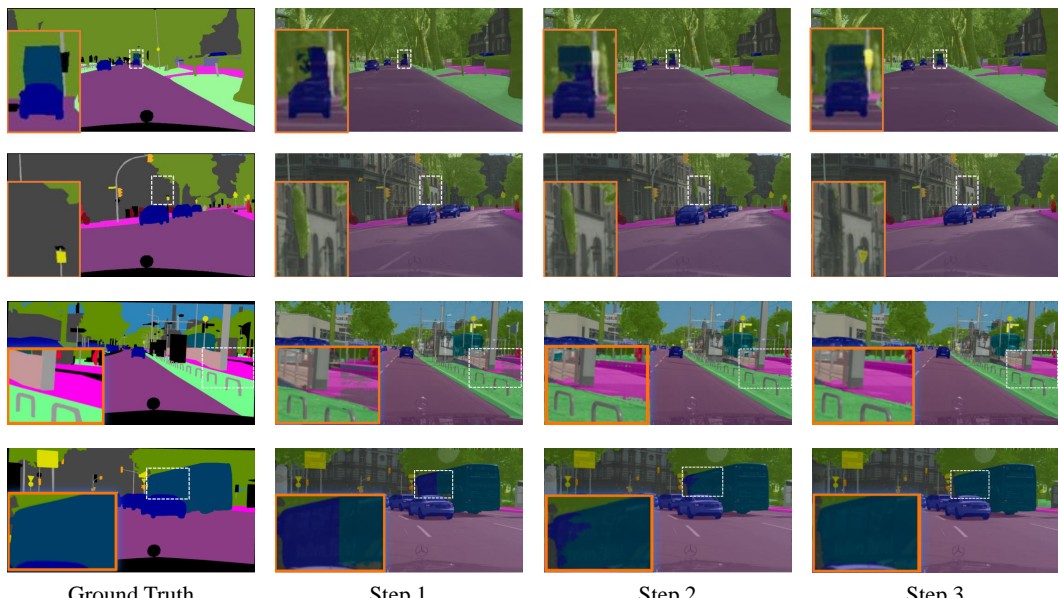

Figure 10: Inference trajectory with diffusion sampling steps. The model gradually refines the prediction, starting from a coarse estimation in Step 1 and progressively improving the results in Step 2. The final output in Step 3 closely resembles the ground truth, demonstrating the effectiveness of DiffMatch in capturing fine-grained details and accurately delineating the boundaries of the changed buildings.

## J    DETAILED ILLUSTRATION OF DIFFMATCH FRAMEWORK

In this section, we provide detailed illustrations for a clearer understanding of the DiffMatch framework and the conditional discrete data generation pipeline using the diffusion process strategy. Figure 11 presents a comprehensive overview of the key components in DiffMatch, including the feature extractor, mask denoiser, and the supervised and unsupervised loss calculations. Figure 12 further illustrates the forward and reverse diffusion processes employed in the conditional discrete data generation pipeline for semi-supervised semantic segmentation.

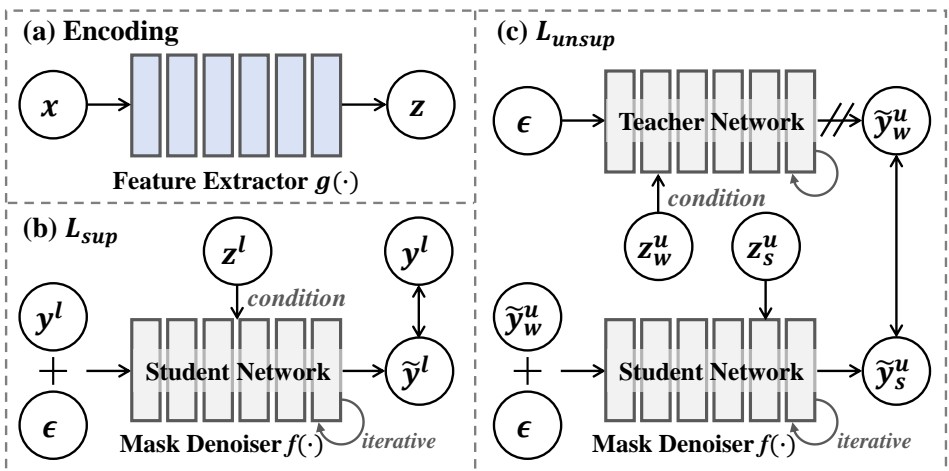

Figure 11: Illustration of the DiffMatch framework. (a) The encoding process. The feature extractor $g(\cdot)$ takes an input image $x$ and outputs the pixel embedding $z$. (b) Supervised loss calculation. The ground truth mask $y^l$ is corrupted with noise $\epsilon$ sampled from the Gaussian distribution to obtain the noisy mask $y_t^l$. The mask denoiser $f(\cdot)$ takes $y_t^l$ and $z^l$ as inputs to predict the denoised mask $\tilde{y}^l$. The supervised loss $\mathcal{L}_{sup}$ is computed between $\tilde{y}^l$ and $y^l$. (c) Unsupervised loss calculation. Weak and strong augmentations are applied to the unlabeled image $\mathbf{x}^u$ to obtain $\mathbf{x}_w^u$ and $\mathbf{x}_s^u$. The teacher network generates pseudo labels $\tilde{y}_{0,w}^u$ by denoising $\epsilon$ conditioned on $z_w^u$. Noise is injected into $\tilde{y}_{0,w}^u$ to obtain $\tilde{y}_{t,w}^u$. The student network denoises $\tilde{y}_{t,w}^u$ conditioned on $z_s^u$ to predict $\tilde{y}_s^u$. The unsupervised loss $\mathcal{L}_{unsup}$ is calculated between $\tilde{y}_s^u$ and $\tilde{y}_{0,w}^u$.

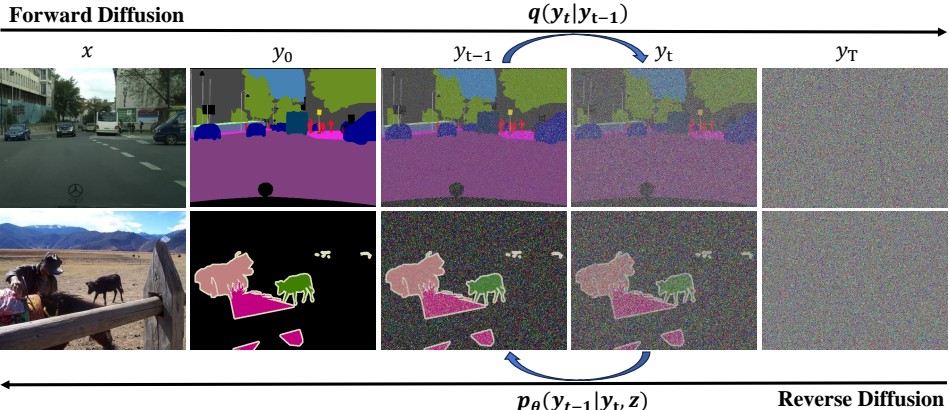

Figure 12: Illustration of the conditional discrete data generation pipeline for semi-supervised semantic segmentation using the diffusion process strategy. The forward diffusion process $q(\boldsymbol{y}_t|\boldsymbol{y}_{t-1})$ progressively corrupts the input mask $\boldsymbol{y}_0$ by adding Gaussian noise at each time step $t$, resulting in the noisy mask $\boldsymbol{y}_t$. The reverse diffusion process $p_\theta(\boldsymbol{y}_{t-1}|\boldsymbol{y}_t, \boldsymbol{z})$ learns to denoise the noisy mask $\boldsymbol{y}_t$ conditioned on the pixel embedding $\boldsymbol{z}$ to recover the mask $\boldsymbol{y}_{t-1}$ at previous time step. The denoising is performed iteratively, with the mask denoiser $f(\cdot)$ predicting the denoised mask at each step.

