# OpenReview forum: "Towards Unbiased Learning in Semi-Supervised Semantic Segmentation"
_ICLR.cc/2025/Conference — ICLR 2025 Poster_

### Official Review · Reviewer_nyBx · 2024-10-17

**Soundness:** 2
**Presentation:** 3
**Contribution:** 2
**Rating:** 6
**Confidence:** 4

**Summary:**

This paper presents DiffMatch, which aims to use Diffution models to solve the data imbalance in SSS. Specifically, Diffmatch employs conditional discrete data generation to align the predictions with real class distributions. The authors conduct extensive experiments to demonstrate their effectiveness. While the paper tells a good story and offers thorough formulations as evidence, there remain certain concerns that could potentially impact the overall quality.

**Strengths:**

1. The motivation is clear and easy to follow.
2. The authors provide extensive experimental results to demonstrate the effectivness of their method.

**Weaknesses:**

Method:
1. Using diffusion models to solve segmentation is not very sound to me, as diffusion models are not yet mainstream in visual perception solutions. Despite mentioning some papers in related works, these approaches do not rank as top performers in segmentation.
2. The primary challenge in SSS is noisy pseudo labels. Even though the DiffMatch can align the predictions of unlabeled data, there is no guarantee they are correct.
3. I think the learning of unlabeled images is still based on the knowledge learned from labeled images. The process of adding noise and denoise did not bring new information to improve the quality of pseudo labels.
4. How to measure the effectiveness of DiffMatch in prediction alignment? Instead of final segmentation results, can we analyze them by distribution visualization or other methods?
5. Intuitively, the diffusion process will increase computation costs. However, as in Table 5, DiffMatch is more efficient than other methods. In addition, it seems that only a few sampling steps can already achieve good performances. Why? In addition, how are the effects of the new loss functions in efficiency (Secs 3.2 and 3.3)?

Experiments:
1. The authors claim that DiffMatch can extend to remote sensing and medical. However, DiffMatch is designed for the long-tailed problem with many categories. WHU-CD is a binary change detection dataset. ACDC is for heart segmentation. Both datasets do not have long-tail problems. Why DiffMatch still outperforms other methods by a clear margin? I think if the authors would like to prove the effectiveness of DiffMatch in long-tail, it would be better to select other datasets (e.g., containing different organs and tumors). Otherwise, such statements may be over-claimed.
2. Some recent works employ transformers and push the SSS to another level. Can DiffMatch apply to transformers?
2. The authors claimed that "DiffMatch will serve as a solid baseline and facilitate future research", but I don't think Diffusion models would be an extendable design for future research in SSS.
3. DiffMatch compares a lot with UniMatch and follows its settings. UniMatch released a strong codebase with checkpoints and training logs for following research. I think this part is important for DiffMatch to be a solid baseline.

Overall, I am incline to the negative aspect currently. I am afraid that potential readers may consider this paper as a story but not a practical solution. However, I would be very glad to improve my ratings if the authors could solve my concerns.

**Questions:**

Please refer to the weakness for the questions.

Minor Suggestion (not vital):
Figs 2 and 3 are not very illustrative and too small. Could they be enlarged, considering their significance in explaining the method? And the caption can be more illustrative since there are numerous mathematical symbols in the figures. It was only after delving into Section 3 that I grasped the workings of DiffMatch. The current presentation may be challenging to follow.

---

### Official Review · Reviewer_w9jN · 2024-11-02

**Soundness:** 3
**Presentation:** 3
**Contribution:** 3
**Rating:** 6
**Confidence:** 3

**Summary:**

The paper introduces a new method called DiffMatch that aims to address the Matthew effect in semi-supervised semantic segmentation. The approach reframes semi-supervised semantic segmentation as a conditional discrete data generation problem, with a debiased adjustment. This adjustment mathematically modifies the conditional reverse probability at each sampling step to produce unbiased results. The experimental results show that DiffMatch outperforms current state-of-the-art methods, especially in scenarios with limited labeling and severe class imbalance issues.

**Strengths:**

DiffMatch is technically innovative, particularly in terms of incorporating debiased adjustments and conditional discrete data generation problems. It is not only theoretically appealing, but also performs well in experiments. Your analysis and explanation of how DiffMatch performs well in both head and tail classes are clear and convincing, which improves the model's ability to recognize minority classes and has potential applications in a variety of computer vision tasks.

**Weaknesses:**

1. MORE VISUALIZATION: I have noticed that Remote Sensing Interpretation and Medical Image segmentation are mentioned in the scalability for other scenarios, I suggest that the authors further provide information on the visualization of the results of the two datasets mentioned below to help readers more intuitively understand DiffMatch's performance.

2. SCALABILITY FOR OTHER SCENARIOS: The experimental part is extensive and in-depth, covering the fields of natural images, remote sensing images and medical images. I suggest that the authors further explore how the model performs on different datasets, and whether there are specific categories or scenarios that have a greater impact on model performance.

3. Writing quality: The overall writing quality of the paper is good, with clear logic and a well-structured layout. However, some sections, such as the discussion of related work, could be further expanded to better position the study within the existing literature.

**Questions:**

Please refer to the weakness section for detailed suggestions.

---

### Official Review · Reviewer_iDj4 · 2024-11-03

**Soundness:** 4
**Presentation:** 3
**Contribution:** 3
**Rating:** 8
**Confidence:** 5

**Summary:**

This paper analyzes the issue of class imbalance, which leads to model predictions being biased towards head classes and away from tail classes, summarizing it as the Matthew effect. To address this problem, the authors propose a new method called DiffMatch, which formulates the semi-supervised semantic segmentation task as a conditional discrete data generation problem, alleviating the Matthew effect from a generative perspective. Furthermore, based on the mathematical derivations, the authors introduce a debiased adjustment to adjust the conditional reverse probability, reducing the risk of overfitting to head classes during the generation process. Experimental results demonstrate the effectiveness of their proposed method in the challenging dense pixel-level classification task.

**Strengths:**

1.	The authors establish a well-thought-out framework to integrate generative models into semi-supervised learning (SSL) for the challenging semi-supervised semantic segmentation task, mitigating the class imbalance issue from an interesting generative perspective, which is quite novel.

2.	The mathematical derivation of the debiased adjustment is ingenious and thoughtful. The theoretical analysis not only explains the reasons behind the proposed method's effectiveness in addressing the class imbalance issue in generative modeling but also provides valuable insights and guidance for future research in SSL techniques.

3.	The paper is presented in a clear and concise manner, effectively conveying the intuitive motivation and ideas behind the proposed method. The authors thoroughly analyze the Matthew effect present in previous methods and leverage mathematical tools to alleviate the issue from a generative perspective.

4.	The evaluation in this paper is sufficient and comprehensive. In addition to evaluating the effectiveness of the proposed method on common semi-supervised semantic segmentation benchmarks, the authors extend their analysis to other domains, including remote sensing and medical images.

**Weaknesses:**

1.	Although the authors report the trade-off between performance and computational cost of the proposed method during inference, the computational cost during the *training* stage compared to the baseline should also be reported to demonstrate the method's effectiveness.

2.	The specific implementation details of the scheme adopted in this work have not been mentioned. Is it a teacher-student network scheme or another type of scheme?

3.	In Appendix C, the factor t is introduced in Equality ③ without sufficient description. Please provide more details on the reasoning behind this step and how t is derived. What is the impact of t on the overall debiased adjustment? Discuss whether removing or simplifying t would impact the method's performance.

4.	While I understand that the ICLR submission deadline is earlier than the release phase for ECCV papers, I encourage the authors to include comparisons with some recent works from ECCV 2024, such as MPMC [a], which could serve as a recent example. [a] Beyond Pixels: Semi-Supervised Semantic Segmentation with a Multi-scale Patch-based Multi-Label Classifier. ECCV 2024.

5.	I encourage the authors to discuss the limitations of the proposed method.

6.	Minor typos: The “its” in “taking advantage of its their better class-imbalance tolerance” in L19 should be deleted.

**Questions:**

There are also some minor questions.

1.	In the experiments, the authors use a fixed number of diffusion sampling steps, and the results demonstrate that the proposed method achieves competitive performance with reasonable computational cost. However, it is worth considering if a dynamic adjustment of sampling steps could potentially lead to further reductions in computational cost. Please discuss this possibility, as I think it will be interesting and feasible.

2.	I am curious about the effectiveness of DiffMatch when applied to other related domains, such as semi-supervised classification.

Overall, I think now this paper meets the bar for ICLR, and I will consider raising my score if my questions are addressed during the rebuttal.

---

### Meta-Review · Area_Chair_FC6E · 2024-12-21

**Metareview:**

This paper addresses semi-supervised semantic segmentation using diffusion models in a teacher-student SSL framework. The authors claim that generative models are more robust to class imbalance and, hence, developed a novel and interesting discrete data generation pipeline for semi-supervised semantic segmentation, which is novel and interesting. The paper is well-written with sufficient details for reproducing experimental results. The proposed pipeline is easy to implement with minor changes to standard diffusion model training. The paper also gave theoretical justification from an information-theoretical perspective for why DiffMatch improves the quality of pseudo labels, which is appreciated by reviewers.
The downside of the method is increased inference cost compared to discriminative models due to the multi-step denoising process in diffusion models. Also, diffusion models are data-driven and have problems when trained with imbalanced data.
The AC recommends this paper for acceptance primarily because it successfully adapts diffusion models for semi-supervised semantic segmentation and provides sound theoretical justification. All three reviewers have positive ratings for this paper.

**Additional Comments On Reviewer Discussion:**

Reviewer nyBx was not sure why a diffusion model was used for this problem, as SOTA models for semantic segmentation are not based on diffusion models.
The authors did not fully address the concern but mentioned that the backbone of diffusion models here is lightweight and similar to DeepLabV3+. Hence, the model architecture is not more complex yet its performance is superior to discriminative models.

---

### Decision · Program_Chairs · 2025-01-22

Accept (Poster)